# TRACING THE TRACES: LATENT TEMPORAL SIGNALS FOR EFFICIENT AND ACCURATE REASONING

**Martina G. Vilas**[1,2*]  **Safoora Yousefi**[2]  **Besmira Nushi**[3†]  **Eric Horvitz**[2]

**Vidhisha Balachandran**[2]

[1]Goethe University Frankfurt   [2]Microsoft Research   [3]NVIDIA

## ABSTRACT

Reasoning models improve their problem-solving ability through inference-time scaling, allocating more compute via longer and multiple token budgets. Identifying which reasoning traces are likely to succeed remains a key opportunity: reliably predicting productive thinking paths can substantially reduce wasted computation and improve overall efficiency. We introduce *Latent-Trajectory* signals that characterize the temporal evolution of a model's internal representations during the generation of intermediate reasoning tokens. By analyzing both the extent and temporal course of latent representational change, as well as its alignment with the final state, we show that these signals are strong predictors of solution accuracy, outperforming conventional output-based confidence measures. We use latent-trajectory signals to guide answer selection across multiple sampled generations, demonstrating that they make test-time scaling more effective and efficient, reducing token usage by up to 70% while preserving and even improving accuracy on average in comparison with majority voting. Finally, we show that these signals often emerge early in the reasoning trace, which enables early selection and allocation of compute to the most promising answer candidates during generation. Our findings contribute not only practical strategies for inference-time efficiency, but also a deeper interpretability perspective on how reasoning processes are represented and differentiated in latent space.

## 1 INTRODUCTION

Recent advances in large language models (LLMs) have shown that complex reasoning tasks can be solved more effectively by scaling computing at inference time to generate longer and multiple chains-of-thought (*reasoning traces*) and aggregating them into a final solution (Guo et al., 2025; Abdin et al., 2025; OpenAI, 2024; Yang et al., 2025a). However, not all reasoning traces are equal: while some contain productive steps that lead to correct answers, others may deviate into unproductive behaviors such as overthinking, failing to converge on a valid solution strategy, or exhibiting inconsistent reasoning (Shojaee et al., 2025; Chen et al., 2024; Sun et al., 2025). Identifying the quality of a reasoning trace (the likelihood of it leading to a correct solution) is critical. It not only enables more reliable prediction of correct answers, but it can also improve computational efficiency by potentially avoiding wasted effort on unproductive paths, and can provide feedback signals that enhance model training. By understanding which reasoning processes are effective, we can systematically guide models toward reinforcing productive strategies and suppressing ineffective ones.

Prior work has approached this problem by inspecting reasoning traces in their surface natural-language form and identifying behaviors that lead to accurate answers, an approach that typically relies on costly human or model annotation strategies (Lee et al., 2025; Gandhi et al., 2025). In addition, natural language traces may not always reflect the underlying strategies that models employ (Chen et al., 2025; Stechly et al., 2025), and some models are trained to produce intermediate latent

---

*Work done during internship at Microsoft Research. Correspondence to martinagvilas@em.uni-frankfurt.de and vidhishab@microsoft.com

†Work done at Microsoft Research

embeddings rather than explicit text (Hao et al., 2024). Thus, language alone may be an unreliable proxy for evaluating reasoning trace quality. Other work has explored heuristic signals like trace length (Hassid et al., 2025; Marjanović et al., 2025), output distribution statistics (Kadavath et al., 2022; Yona et al., 2022), agreement-based self-consistency (Wang et al., 2023), or using trained verifiers (Li et al., 2023; Zhang et al., 2024) to identify correct solutions, but these methods often trade accuracy for simplicity, or computational cost for accuracy.

In this work, we explore an alternative direction that solely leverages a model's trajectory of hidden states to predict which traces lead to a correct solution. Previous studies have shown that probing hidden states can reveal informative signals about safety (Turner et al., 2023; Zou et al., 2023), learning dynamics (Olsson et al., 2022; Hosseini & Fedorenko, 2023), reliability (Meng et al., 2022; Yuksekgonul et al., 2024), and performance (Wang et al., 2024) of LLMs. Building on this perspective, we hypothesize that the temporal evolution of hidden states during the generation of intermediate reasoning tokens contains predictive information about the final solution correctness, and can be leveraged for more compute-efficient and accurate inference.

We introduce a family of *Latent-Trajectory* (LT) signals that capture three complementary temporal aspects of a model's internal representational trajectory (see Figure 1): (i) the total representational change from the start to the end of the trace, (ii) the cumulative change taken along the way, and (iii) the extent to which the intermediate changes progress towards or away from the final state. These metrics operate directly on hidden states, require no additional training or external annotations, and can be computed during inference.

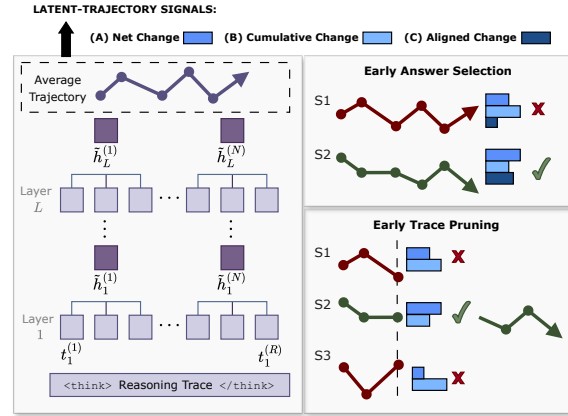

Figure 1: Latent-trajectory framework. Trajectory vectors are constructed from token-level hidden states, and a set of three signals is derived to quantify their temporal evolution. These signals predict successful traces and enable answer selection and early path selection in multi-sample inference.

Our experiments evaluate the use of LT signals across families of reasoning-enabled LLMs (DeepSeek-R1-Distill-Qwen14B, Phi4-Reasoning-Plus, Qwen3-14B) and domains spanning science, math, and path optimization problems. We show that LT can reliably distinguish between traces leading to correct versus incorrect answers, yielding significantly higher discriminatory power than methods using other model internal or output-distribution-based signals. In addition, we demonstrate that LT can be leveraged during inference to achieve both higher efficiency and improved accuracy. In sample scaling experiments, early answer selection using LT yields up to a 70% reduction in token usage, along with 2.5% average accuracy gain over majority-vote baselines by reducing the number of generations sampled. Finally, we show that these signals manifest in the early stages of the reasoning trace, allowing for early identification of promising candidates and targeted compute allocation.

Overall, our results show that a model's internal dynamics can be reliable predictors of reasoning quality, offering both practical tools for inference-time control and interpretability insights into how reasoning trajectories evolve, opening paths for broader applications that exploit internal signals for efficiency, accuracy, and calibrated decision-making.

## 2 RELATED WORK

**Assessing Reasoning Quality:** A growing body of work seeks to quantify the quality of reasoning traces in order to predict solution accuracy with high reliability. Many strategies involve employing verifier models, external or self, to assess the correctness of candidate answers (Weng et al., 2023; Madaan et al., 2023; Zhang et al., 2024). These approaches are effective but substantially increase inference cost. An alternative direction performs fine-grained analyses of the trace surface form, proposing metrics that target factual and logical validity, as well as linguistic and semantic coherence (Wu et al., 2025; Golovneva et al., 2022). Heuristics derived from output token distributions or from

trace length have also been explored to decide whether a path is likely to be accurate (Hassid et al., 2025; Kadavath et al., 2022; Yona et al., 2022). Such methods often require annotation or structured extraction from traces, which introduces dependence on human raters or auxiliary expert models and can lead to model-specific heuristics. In contrast, LT signals are computed directly at inference time without a teacher model or additional runs, which yields a more efficient procedure. Concurrent work trains model-specific probes over hidden representations to detect when intermediate answers are likely correct (Zhang et al., 2025). Our approach shares the objective but remains training-free and can be applied to diverse models and datasets with minimal setup.

**Representational Analysis:** Previous studies have shown that probing an LLM's hidden states reveals informative signals about reliability (e.g. Meng et al., 2022; Yuksekgonul et al., 2024), safety (e.g. Turner et al., 2023; Zou et al., 2023), performance (e.g. Herrmann et al., 2025; Wang et al., 2024), and learning dynamics (e.g. Olsson et al., 2022; Hosseini & Fedorenko, 2023). We extend this research direction to leverage hidden states to predict accuracy in reasoning models.

Recent efforts have demonstrated that information-theoretic, geometric, and invariance-based metrics capture intermediate representations that are more predictive of downstream performance than later layers (Skean et al., 2025). Closest to our approach, Wang et al. (2024) examines representational curvature across layers (i.e. spatial perspective) for predicting answer accuracy in instruction-tuned models. We differ by adopting a temporal perspective across tokens and focusing specifically on reasoning models. Concurrent work (Li et al., 2025) extends sequential representational analysis to detect repetition loops in mathematical reasoning, further supporting the premise that temporal latent dynamics provide valuable insight into model behavior.

**Efficient Inference Scaling:** Scaling up inference-time computation is a key factor in improving reasoning performance in LLMs (OpenAI, 2024; Guo et al., 2025; Abdin et al., 2025). While effective, previous studies (Balachandran et al., 2025; Shojaee et al., 2025; Sui et al., 2025) show that models trained to generate long reasoning traces exhibit 'overthinking' and consume compute even after reaching a correct solution. This has motivated efforts to curb such behavior, either by training models to produce more concise reasoning (Kang et al., 2025a; Shrivastava et al., 2025) or by dynamically halting trace generation once the model is confident in its answer (Yang et al., 2025b; Zhang et al., 2025). Another inference-time scaling strategy is to generate multiple samples and aggregate answers using self-consistency (Wang et al., 2023), external verifiers (Zhang et al., 2024), or iterative self-verification (Madaan et al., 2023; Balachandran et al., 2025). These methods boost accuracy for both standard and reasoning models but come with substantially higher computational cost. Recent work has sought to improve efficiency by pruning reasoning paths with trained classifiers (Manvi et al., 2024; Li et al., 2024). In contrast, our experiments show that Latent-Trajectory Signals provide a training-free way to guide reasoning-path selection and answer aggregation.

## 3 LATENT-TRAJECTORY SIGNALS OF REASONING QUALITY

Given a problem, reasoning models generate a sequence of tokens composed of a reasoning trace followed by a final answer. The trace is often delimited by special tokens ($\{trace\_start\}$, $\{trace\_end\}$), such that:

$$q_1, \ldots, q_i \, \{\texttt{trace\_start}\} \, t_1, \ldots, t_r \, \{\texttt{trace\_end}\} \, a_1, \ldots, a_j,$$

where $q_1, \ldots, q_i$ are the user query (problem) tokens, $t_1, \ldots, t_r$ are the reasoning trace tokens, and $a_1, \ldots, a_j$ are the final answer tokens. For each position $r \in \{1, \ldots, R\}$ within the reasoning trace, the model produces a hidden state of activations at each layer $l \in \{1, \ldots, L\}$, denoted by $h_l^{(r)} \in \mathbb{R}^d$. These hidden states form a 2D array of $d$-sized representations, indexed by layer and token position, and encode the *latent space* of the model at each step of the reasoning trace (see Figure 1).

### 3.1 LATENT-TRAJECTORY SIGNALS

We seek to characterize the quality of a model's intermediate reasoning by tracking the evolution of its internal representations through the trace. To quantify this, we average token-level activations into a sequence of segment-level states and extract trajectory signals that quantify the magnitude and geometry of representational change.

First, to enhance the robustness of the signal and reduce dimensionality, we divide the reasoning trace $t_1, \ldots, t_r$ into non-overlapping *reasoning segments*, where each segment is a contiguous block of $k$ tokens ($k = 500$)[1]. For each transformer layer $l \in \{1, \ldots, L\}$ and segment index $n \in \{1, \ldots, N\}$, we compute the segment-level hidden state $\tilde{h}_l^{(n)}$ by averaging the token hidden states within that segment. Intuitively, $\tilde{h}_l^{(n)}$ corresponds to the average representation the model maintains in latent space while processing segment $n$. This temporal coarse-graining smooths local fluctuations in token-level dynamics while preserving the large-scale evolution of the model's latent space over the trace. The sequence $\{\tilde{h}_l^{(1)}, \ldots, \tilde{h}_l^{(N)}\}$ at layer $l$ provides a trajectory-level encoding of the hidden-state evolution over the intermediate reasoning tokens.

Given the segment hidden states, we define two basic vectors that anchor our trajectory signals. The *reasoning drift vector*: $u_l = \tilde{h}_l^{(N)} - \tilde{h}_l^{(1)}$, captures the overall direction and distance the model's internal state travels during the trace. Complementarily, the *update vector* for segment $n$: $v_l^{(n)} = \tilde{h}_l^{(n)} - \tilde{h}_l^{(n-1)}$, $n = 2, \ldots, N$ describes the incremental change between consecutive reasoning segments. Taken together, $u_l$ and $v_l^{(n)}$ capture not only the overall extent of representational movement, but also the step-by-step dynamics of how that movement unfolds.

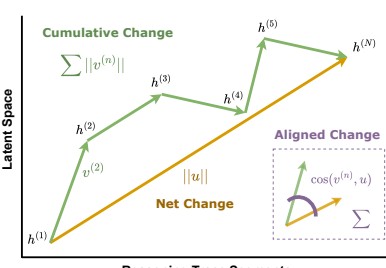

Figure 2: Latent-trajectory signals.

From these primitives we derive three complementary signals that summarize (i) overall representational change, (ii) accumulated change over the trace, and (iii) extent of progress towards the final state (see Figure 2). Each signal is computed per layer and then averaged across layers to yield a single score per trace.

**Net Change.** First, we explore whether intermediate reasoning substantially alters the model's latent space and whether such change is predictive of accuracy. To assess this, we measure the magnitude of representational change in the latent space between the first and last reasoning segment.

Formally, we measure the norm of the drift vector $u_l$ at each layer, which encodes the magnitude of this change, and normalize by the number of segments to control for trace length. Finally, we average across layers to obtain a single score:

$$\text{NETCHANGE} = \frac{1}{L} \sum_{l \in L} \frac{\|u_l\|_2}{N}$$

Larger values indicate that the final hidden state has substantially changed from the initial state, suggesting that the reasoning steps produced a significant overall change in representational space.

**Cumulative Change.** While Net Change measures the overall representational change between the initial and final reasoning segments, it does not characterize the intermediate latent-space changes. To summarize the total amount of representational movement along the trace, we additionally compute the cumulative magnitude of the sequential updates to the reasoning trace.

We consider the update vectors $v_l^{(n)}$, which represent the changes in layer $l$ between consecutive reasoning segments. The norm of the update vectors $\|v_l^{(n)}\|_2$ gives the magnitude of change at each step, and aggregating the norms across all segments captures the total movement along the trajectory. Finally, averaging across layers yields a single score:

$$\text{CUMULATIVECHANGE} = \frac{1}{L} \sum_{l \in L} \sum_{n=2}^{N} \|v_l^{(n)}\|_2$$

Intuitively, Cumulative Change quantifies the overall shifts in representations during the course of reasoning, independent of the final states. Larger values encode significant variations in representations across segments.

---

[1]We experimented with various segmentation methods, including delimiters. See Appendix J.

**Aligned Change.** Beyond measuring the magnitudes of overall and intermediate changes, we ask whether intermediate updates tend to proceed in the same direction as the final outcome. We hypothesize that for reasoning traces leading to accurate solutions, the sequence of updates should mostly advance toward the final representation.

Formally, this is assessed by comparing each update vector $v_l^{(n)}$ with the drift vector $u_l$. The cosine similarity $\frac{\langle v_l^{(n)}, u_l \rangle}{||v_l^{(n)}||_2 ||u_l||_2}$ measures the angle between the two, indicating whether each local update proceeds in the same general direction as the overall displacement. Averaging across segments and layers yields a single score:

$$\text{ALIGNEDCHANGE} = \frac{1}{L} \sum_{l \in L} \frac{1}{N-1} \sum_{n=2}^{N} \frac{\langle v_l^{(n)}, u_l \rangle}{\|v_l^{(n)}\|_2 \|u_l\|_2}.$$

Higher values suggest that intermediate updates are aligned with the overall progress toward the final state, while lower values indicate they are inconsistent or even opposed to it.

## 4 EXPERIMENTAL SETUP

### 4.1 BASELINES

We use two alternative approaches as baselines: (1) *Cross-Layer Signals*, which summarize representational changes *across layers* within a segment, and (2) *Output Distribution Measures*, which estimate confidence from the token distribution at the final answer.

**Cross-Layer Signals:** Previous work has shown that changes across layers can be predictive of answer accuracy in CoT. Following (Wang et al., 2024), for each reasoning segment $n$, we compute the mean *magnitude* and *angle* of layer-to-layer changes and then average over segments:

$$\text{LAYERMAG}^{(n)} = \frac{1}{L} \sum_{l=2}^{L} \frac{\|\tilde{h}_l - \tilde{h}_{l-1}\|_2}{\|\tilde{h}_L - \tilde{h}_1\|_2}; \quad \text{LAYERANG}^{(n)} = \frac{1}{L} \sum_{l=2}^{L} \frac{\arccos(\cos(\tilde{h}_l, \tilde{h}_{l-1}))}{\arccos(\cos(\tilde{h}_L, \tilde{h}_1))}$$

**Output Distribution Measures:** Output distribution–based measures are commonly used as estimates of model confidence (Yona et al., 2022; Kadavath et al., 2022; Manakul et al., 2023), and can be used as proxies for final answer reliability. To compare against these metrics, we elicit the final answer post the reasoning trace end using prompts of the form $[\ldots \{trace\_end\}$ Final Answer:$]$, and examine the probability distribution over the token that follows. Based on findings from Yona et al. (2022), we considered three best performing output distribution measures: (i) **Logit Margin**: the difference between the top-two token logits; (ii) **Entropy**: the entropy of the token distribution. (iii) **Perplexity**: computed as the inverse probability of the model's top-ranked token, providing a scalar measure of confidence in its most likely continuation.

### 4.2 MODELS AND DATASETS

For our main experiments, we evaluate three open-source reasoning models: DeepSeek-R1-Distill-Qwen-14B (R1-D) (Guo et al., 2025), Phi-4 Reasoning Plus Model (PHI4R+) (Abdin et al., 2025), and Qwen3-14B (QWEN3) (Yang et al., 2025a). Our study tests our LT signals across three distinct reasoning domains: (i) *Scientific*, measured using the *GPQA Diamond* benchmark, which comprises 198 graduate-level multiple-choice questions in biology, chemistry, and physics (Rein et al., 2024); (ii) *Mathematical*, evaluated on *AIME 2025*, a 30-problem set from the American Invitational Mathematics Examination (AIME, 2025); (iii) *Algorithmic*, assessed with a stratified subsample ($n = 180$) of the *TSP* benchmark, consisting of path-optimization problems across varying levels of difficulty (graphs of 6 to 13 nodes) (TSP, 2025).

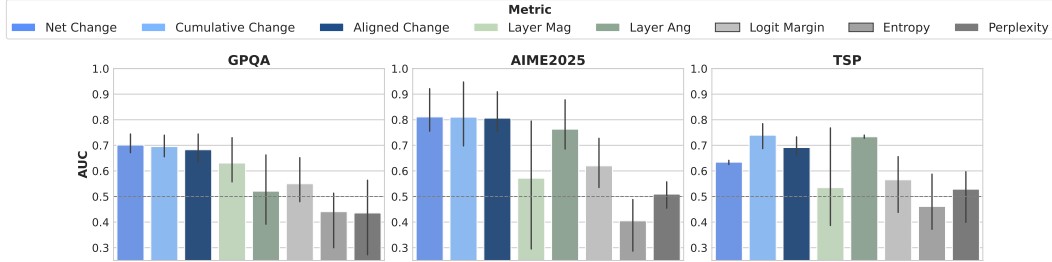

Figure 4: ROC-AUC for distinguishing correct from incorrect predictions using Latent-Trajectory (LT) and baseline metrics. Higher values indicate better discriminative power. For comparability, Cumulative Change and Layer Angle metrics were sign-reversed. LT signals consistently achieve above chance (dashed line) and more reliable discrimination than baseline metrics. Error bars denote variability across models.

## 5 RESULTS

### 5.1 LATENT-TRAJECTORY SIGNALS ARE PREDICTIVE OF SOLUTION ACCURACY

To assess whether LT signals predict solution correctness, we evaluate their discriminative power using the area under the ROC curve (AUC). For each problem, we generate five independent solutions (reasoning trace and final answer). For each trace, we compute the LT scores from the hidden states of the intermediate reasoning tokens. We then compute the ROC-AUC of the signal with respect to accuracy by sweeping a decision threshold over the scores, which captures how well the signals discriminate solution correctness.

We found that Cumulative Change was negatively correlated with accuracy (Spearman's $r = -.38$), which indicates that traces that traverse greater total distance in representation space tend to be less likely to produce correct answers. This finding mechanistically grounds prior behavioral observations that long but highly varying reasoning traces are associated with lower accuracy (Balachandran et al., 2025; Shojaee et al., 2025). In contrast, Net and

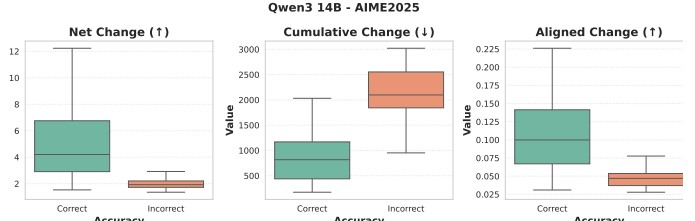

Figure 3: Latent-trajectory signal distributions by accuracy for Qwen3-14B on the AIME 2025 dataset. Correct traces show larger Net/Aligned Change and smaller Cumulative Change than incorrect ones. This indicates that correct reasoning corresponds to larger, more directed representational shifts, while incorrect reasoning involves more wandering and less aligned trajectories.

Aligned Change show positive associations with accuracy (Net Change $r = .28$; Aligned Change $r = .32$). Larger overall representational change from the initial to the final hidden state is therefore linked to better performance, and representational updates that progress more directly toward the final state show an even stronger association. Figure 3 shows the distributions of the three trajectory metrics for Qwen3 on AIME2025. The distribution of values further supports our findings: successful trajectories cover greater distances in latent space, advance more directly toward the final state at intermediate steps, and involve less path deviations. Equivalent plots for each model and dataset are in Appendix A, including plots of layer-wise values for each LT signal.

As shown in Figure 4, LT signals significantly distinguish between reasoning traces that lead to accurate versus inaccurate solutions. *Across datasets, the ROC-AUCs of our three LT signals remain consistently above chance, demonstrating robust predictive power* (Net Change mean ROC-AUC $= 0.71 \pm 0.09$; signed-reversed Cumulative Change $= 0.74 \pm 0.09$; Aligned Change $= 0.73 \pm 0.08$). In contrast, the cross-layer magnitude and angle signals are less reliable and vary substantially across models and reasoning domains (Cross-Layer Magnitude Change $= 0.58 \pm 0.17$; signed-reversed Cross-Layer Angle Change $= 0.67 \pm 0.14$). Output-distribution–based metrics are significantly weaker and less consistent, with performance often close to or below chance level (Logit Margin $= 0.59 \pm 0.10$; Entropy $= 0.44 \pm 0.10$, Perplexity $= 0.49 \pm 0.12$). In summary, our results show

that for models that produce long intermediate traces, signals that capture the temporal evolution in latent space are stronger and more robust predictors of solution accuracy than cross-layer geometry or output-distribution-based confidence measures (see Appendix B for ROC-AUC scores and correlation values for each model-dataset combination).

To more comprehensively assess the predictivity and robustness of the LT metrics, Appendix B broadens the evaluation to include additional models of distinct sizes and architectures (DeepSeek-R1-Llama 8B; DeepSeek-R1-Qwen 32B; DeepSeek-R1-Llama 70B) and a suite of reasoning problems that entail divergent reasoning, including commonsense, creative, narrative, and socially grounded tasks (Srivastava et al., 2023). We observe that our LT signals remain significantly predictive of answer correctness across the new models and reasoning domains. The ROC-AUC values for Net, Cumulative, and Aligned Change remain well above chance and are comparable to those reported for the main benchmarks (Net Change mean ROC-AUC $= 0.71 \pm 0.09$; Cumulative Change $= 0.72 \pm 0.09$; Aligned Change $= 0.70 \pm 0.10$).

## 5.2 LATENT-TRAJECTORY SIGNALS IMPROVE EFFICIENCY AND RELIABILITY OF MULTI-SAMPLE INFERENCE

Building on our previous finding that LT signals strongly predict solution accuracy, we now investigate whether they can guide more accurate and efficient scaling strategies for sampling, selecting, and aggregating solutions in multi-sample inference systems. Previous work demonstrates that generating multiple samples and aggregating them through self-consistency improves both accuracy and reliability in language models (Wang et al., 2023; Kang et al., 2025b). In practice, majority voting (MV) has become the default approach for recent releases of reasoning models (Abdin et al., 2025; Guo et al., 2025), since a single inference pass is rarely sufficient for robust performance, especially in applications or agentic settings (Besta et al., 2025). This robustness, however, comes with increased inference costs, particularly for reasoning models, where long chains of thought lead token usage to grow by an order of magnitude with each additional sample. Here, we examine whether LT-based selection can preserve the benefits of MV while reducing sample and token budget.

**Experiment Setup:** We generate multiple samples sequentially from the model and use LT signals to decide online whether the current trace is likely correct and should be used as the final answer, or whether additional samples are needed. Once a signal exceeds a calibrated threshold, we accept the solution early and stop sampling. If no samples cross the threshold after at most $k$ attempts, we fall back to MV over the collected candidates (see Figure 5). This allows datapoints with strong internal signals to be resolved quickly with fewer samples, while datapoints with weaker signals rely on the robustness of aggregation. We set $k = 5$ and repeat this procedure independently for each signal.

We compared our approach against two sample aggregation baselines: (i) MV, and (ii) shortest-answer selection, which chooses out of the sampled answers the candidate with the fewest tokens, motivated by recent findings that shorter completions are strong signals of accuracy (Hassid et al., 2025; Shrivastava et al., 2025; Marjanović et al., 2025).

We select decision thresholds $\tau$ using a cross-validation approach (see Appendix C for details). On the calibration set, we form candidate thresholds from quantiles of the metric among incorrect solutions, so each candidate fixes the proportion of incorrect solutions that lie beyond the cut-off. For each candidate, we simulated the full decision rule on the calibration subset, accepting a solution early when the signal crosses the threshold and otherwise aggregating with MV. The best-calibrated threshold is then evaluated on the remaining data. Appendix G further explores a global threshold selection strategy that obtains similar results as dataset-specific approaches.

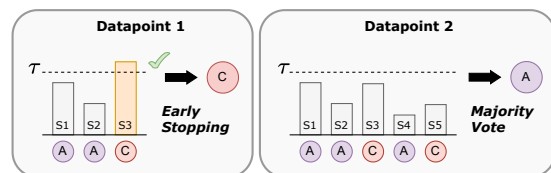

Figure 5: Candidate solutions for a problem are evaluated sequentially. If a solution's signal value exceeds $\tau$, it is immediately accepted as the final prediction. If no solution crosses $\tau$, the final answer is chosen via MV.

We report accuracy as the fraction of problems solved correctly, and efficiency in terms of (i) the average number of samples required, and (ii) the proportion of reasoning tokens consumed relative

to running the full inference procedure with five samples. Reported results are averaged over the splits.

In addition to exploring each metric separately, we built a **Combined LT score** from a weighted sum of the LT values, where signals that are more strongly associated with accuracy on the calibration set contributed more to the final score (Appendix D reports details on its construction).

**Results:** As shown in Table 1, LT signals improve both efficiency and accuracy relative to self-consistency based majority vote. On GPQA, R1-D gains about $2\%$, Qwen3 remains stable, and Phi4R+ maintains competitive accuracy. On AIME2025, improvements are more pronounced with $4\%$ for R1-D, $2\%$ for Phi4R+, and a substantial $12\%$ for Qwen3. On TSP, all models benefit, with consistent gains of $1$–$3\%$. These results show that LT thresholds not only preserve correctness across settings, but often deliver meaningful boosts by identifying correct reasoning paths even when the majority of solutions are incorrect. Efficiency gains are considerably larger. On R1-D, LT signals reduce the average number of tokens required to match or outperform MV by $50$–$66\%$ across datasets, Qwen3 achieves reductions of about $50$-$55\%$, and Phi4R+ reduces sampling by $30$–$35\%$. Across all settings, the Shortest@5 baseline reduces accuracy by an average of $1.4\%$, showing that length alone is an unreliable proxy for correctness. In Appendix E, we further report that LT signals trigger early answer selection for $>85\%$ of data points on average across datasets, and that accuracy within this subset consistently exceeds the baselines, confirming that LT signals concentrate probability mass on more reliable solutions.

Overall, *Latent-Trajectory-guided selection preserves, and often improves, the reliability of majority-vote aggregation while substantially reducing inference costs*, offering a reliable alternative to fixed-sample aggregation. The Combined LT score is frequently competitive with the best individual signal and, in most cases, cuts token usage by at least half. This makes it a practical and effective choice when applied to different models and datasets. At an aggregate level, compared to MV@5, LT strategies exhibit: (i) *Sample savings*: Number of sampled answers is reduced on average by **58%** ($32.0$–$75.6\%$); (ii) *Token savings*: As a consequence of sample savings, token usage (and thereby inference cost) is reduced on average by **48%** ($14.5$–$70.6\%$); (iii) *Accuracy improvement*: Accuracy increases on average by **2.46%** ($-1.4$–$14.1\%$).

Table 1: Accuracy and efficiency with Latent-Trajectory (LT) signals. Baselines are MV@5 (majority vote across 5 samples) and Shortest@5 (shortest of 5 samples). Accuracy is reported as a percentage, with parentheses indicating the change relative to MV@5. For efficiency, we report the average number of samples required per datapoint, with parentheses showing the percentage reduction in total token usage relative to MV. **Bold** and **Bold** denote the best and second-best results within each group. ✓ marks cases where the average number of samples was reduced more than half.

| Model | Strategy | GPQA | | AIME2025 | | TSP | |
|---|---|---|---|---|---|---|---|
| | | Acc. (avg % / ΔAcc) | Samples (avg / ΔTok %) | Acc. (avg % / ΔAcc) | Samples (avg / ΔTok %) | Acc. (avg % / ΔAcc) | Samples (avg / ΔTok %) |
| R1-D | MV@5 | 59.90 | 5.00 | 56.67 | 5.00 | 27.50 | 5.00 |
| | Shortest@5 | 60.91 (+1.0) | 5.00 (0) | 50.00 (-6.7) | 5.00 (0) | 28.75 (+1.3) | 5.00 (0) |
| | LT – Net | 61.10 (+1.2) | **1.69** (+53.9) ✓ | **61.90** (+5.2) | **1.22** (+68.7) ✓ | 28.60 (+1.1) | **1.43** (+70.6) ✓ |
| | LT – Cumulative | **62.10** (+2.2) | 1.88 (+48.1) ✓ | 58.70 (+2.0) | 2.56 (+29.9) | **30.90** (+3.4) | **1.61** (+66.4) ✓ |
| | LT – Aligned | 61.10 (+1.2) | **1.58** (+57.0) ✓ | 60.30 (+3.6) | 1.43 (+61.3) ✓ | 29.50 (+2.0) | 2.08 (+57.2) ✓ |
| | LT – Combined | **61.80** (+1.9) | 1.89 (+47.3) ✓ | **61.90** (+5.2) | 2.06 (+43.9) ✓ | **30.10** (+2.6) | **1.43** (+70.3) ✓ |
| Phi4R+ | MV@5 | **70.20** | 5.00 | 80.00 | 5.00 | 41.25 | 5.00 |
| | Shortest@5 | 69.19 (-1.0) | 5.00 (0) | 70.00 (-10.0) | 5.00 (0) | 38.75 (-2.5) | 5.00 (0) |
| | LT – Net | 68.80 (-1.4) | **2.97** (+20.2) | 79.40 (-0.6) | **2.19** (+41.1) ✓ | 42.30 (+1.1) | **1.59** (+67.2) ✓ |
| | LT – Cumulative | **69.60** (-0.6) | **2.99** (+18.9) | 81.00 (+1.0) | 2.43 (+42.2) | **44.40** (+3.1) | 2.63 (+42.2) |
| | LT – Aligned | **69.60** (-0.6) | 3.40 (+14.5) | 82.50 (+2.5) | 2.51 (+30.8) | **44.10** (+2.9) | **1.96** (+58.7) ✓ |
| | LT – Combined | **69.60** (-0.6) | 3.28 (+16.3) | **82.60** (+2.6) | 2.54 (+28.7) | 43.80 (+2.6) | 2.30 (+50.5) ✓ |
| Qwen3 | MV@5 | 63.96 | 5.00 | 70.00 | 5.00 | 36.25 | 5.00 |
| | Shortest@5 | **64.47** (+0.5) | 5.00 (0) | 80.00 (+10.0) | 5.00 (0) | 30.63 (-5.6) | 5.00 (0) |
| | LT – Net | 63.70 (-0.3) | **1.42** (+63.9) ✓ | 79.40 (+9.4) | 1.60 (+57.3) ✓ | 35.40 (-0.9) | 3.18 (+34.0) |
| | LT – Cumulative | 63.30 (-0.7) | 2.25 (+41.3) ✓ | **84.10** (+14.1) | 2.03 (+43.2) ✓ | 36.30 (+0.1) | **1.64** (+65.5) ✓ |
| | LT – Aligned | **64.20** (+0.2) | 1.75 (+52.0) ✓ | 80.90 (+10.9) | **1.59** (+58.2) ✓ | **37.80** (+1.6) | 2.08 (+56.4) ✓ |
| | LT – Combined | 63.70 (-0.3) | 1.70 (+53.4) ✓ | 80.90 (+10.9) | **1.49** (+60.3) ✓ | 36.00 (-0.3) | 2.46 (+48.4) ✓ |

## 5.3 Latent-Trajectory signals enable early selection of high-quality traces

**Experiment Setup:** While the previous section focused on using LT signals for end-of-trace answer selection, we now ask whether these signals can also identify higher-quality trajectories early

in the reasoning process. To investigate this, we run a step-wise incremental early-exit evaluation. We evaluate signals on *partial* traces taken at 500-token intervals up to the full trace. At each checkpoint, we recompute Net Change and Cumulative Change[2] using only the tokens available so far. For each partial trace, the most recent segment is used as the final segment. We then compute the ROC-AUC of each signal at every checkpoint, revealing how predictive power evolves as the trace unfolds, and whether prediction of solution correctness is possible without observing the full trajectory.

To investigate whether these early signals in the reasoning trace can be leveraged during inference, we implement an early path selection policy when sampling multiple generations in parallel. We compute the LT signals on the partial traces and use them as features for a lightweight random forest classifier trained to predict correctness. The classifier selects a single candidate trajectory to continue, while the other four paths are terminated. The chosen trace is decoded to completion, and we report both the accuracy of this early path selection policy and the proportion of tokens saved compared to running all five trajectories to the end. We also compare the performance of our method against that of Wang et al. (2025) (ST-BoN), which selects the sample to be completed until termination based on sample pairwise distances of cross-layer signals.

**Results:** As Figure 6 shows, Net and Cumulative change provide predictive signals well above chance, with ROC-AUC generally increasing as additional tokens are observed. ROC-AUC values above .6 can be obtained within the first 4k tokens, with Net Change being a better predictor than Cumulative Change early in the trace for GPQA and AIME2025. This pattern, however, reverses for TSP, where Cumulative Change is significantly more predictive than Net Change throughout the early and mid trace.

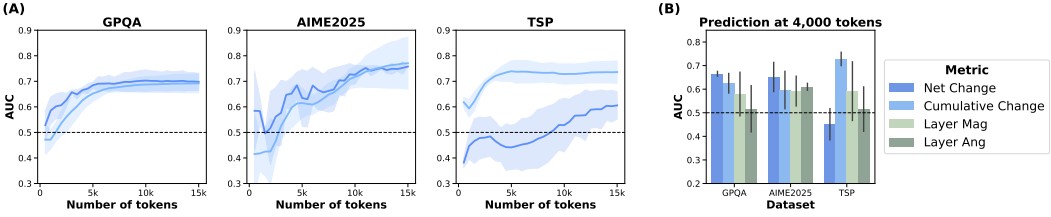

Figure 6: **(A)** Predictive performance (ROC-AUC) of Net and Cumulative Change signals as a function of the number of tokens, across datasets. Shaded regions represent variation across models. **(B)** Comparison of predictive performance at 4k tokens. Error bands indicate variation across models. Performance of LT signals rises early well above the baseline.

Table 2 shows that early selection of high quality paths with LT at 2k tokens reduces inference cost while maintaining or improving accuracy. Across models and datasets, accuracy remains highly competitive with MV@5. R1-D achieves a 6.7% gain on AIME2025 with competitive performance on GPQA and TSP. PhiR+ consistently improves accuracy by 2–4% across datasets, while Qwen3 yields gains of 2–3%. Efficiency gains are more significant, with average token usage reduced by 54% for R1-D, 62% for Qwen3, and 68% for PhiR+. At an aggregate level, we observe: (i) accuracy increases on average by **2.49%**; (ii) token usage reduction on average by **61.2%**. Appendix I further demonstrates that these results are consistent across early point checkpoint selection. Overall, these findings demonstrate that *LT signals can effectively guide early selection of high-quality paths, allocating compute to the most promising generation candidate, thereby achieving competitive accuracy to five-sample majority vote with less than half of the inference cost.*

In addition, we also found that across nearly all model–dataset settings, LT-based early selection matches or exceeds the accuracy of ST-BoN while consistently delivering comparable or greater token savings. LT's deviations are smaller in comparison and its improvements are more uniform, frequently achieving the highest accuracy. Across models and datasets, LT delivers more than double the average accuracy gain of ST-BoN (+2.48% vs +1.12%), at the same rate of token savings.

---

[2]As Aligned Change compares the direction of each segment with respect to the last segment, it is inconsistent when applied earlier in the trace.

Importantly, LT remains computationally cheaper, since it operates on the latent trajectory of a single sample rather than computing pairwise distances across multiple samples as required by ST-BoN.

Table 2: Evaluation of early path selection (at 2k tokens) using LT (LT) signals. Accuracy (%) and Saved Tokens (%) with $\Delta$ relative to Majority Vote (Maj@5).

| Model | Strategy | GPQA | | AIME2025 | | TSP | |
|---|---|---|---|---|---|---|---|
| | | Accuracy (% / $\Delta$ %) | Saved Tokens ($\Delta$ %) | Accuracy (% / $\Delta$ %) | Saved Tokens ($\Delta$ %) | Accuracy (% / $\Delta$ %) | Saved Tokens ($\Delta$ %) |
| R1-D | Maj@5 | 59.90 | - | 56.67 | - | 27.50 | - |
| | ST-BoN | 58.88 (-1.02) | +45.3 | 63.33 (+6.7) | +49.7 | **28.75** (+1.2) | +62.8 |
| | LT | **59.39** (-0.5) | +48.9 | 63.33 (+6.7) | +50.1 | 26.25 (-1.3) | +62.5 |
| Phi4R+ | Maj@5 | 70.20 | - | 80.00 | - | 41.25 | - |
| | ST-BoN | 66.16 (-4.0) | +62.3 | 76.67 (-3.3) | +67.4 | 42.50 (+1.25) | +71.4 |
| | LT | **72.22** (+2.0) | +64.7 | **83.33** (+3.3) | +67.3 | **45.63** (+4.4) | +71.7 |
| Qwen3 | Maj@5 | 63.96 | - | 70.00 | - | 36.25 | - |
| | ST-BoN | 63.63 (+2.54) | +46.8 | **76.67** (+6.7) | +67.4 | 36.25 (+0.0) | +65.8 |
| | LT | **66.50** (+2.5) | +51.0 | 73.33 (+3.3) | +69.1 | **38.13** (+1.9) | +65.7 |

## 6 CONCLUSIONS

Our work introduced a family of LT signals that capture the temporal evolution of reasoning traces within a model's latent space. Across multiple reasoning domains and models, LT metrics predict final-answer correctness significantly above chance and outperform other internal and output-distribution-based baselines. We further demonstrated their utility in practical test-time policies. In inference-scaling experiments, using these signals for answer selection or early path selection reduced token usage and often improved accuracy with respect to strong baselines such as majority vote. Our efficiency gains address two complementary sources of inefficiency: (i) reducing the number of samples required for reliable reasoning, and (ii) shortening individual trajectories by detecting early answers of higher-quality. The approach is model-agnostic, simple to calibrate, and compatible with existing sampling strategies. In addition to practical benefits, our results shed light on the structure of reasoning in latent space, revealing how trajectories unfold during inference and what distinguishes successful from unsuccessful reasoning paths.

There are several opportunities for future work. While we show the real-world utility of these signals at inference time, trajectory-level signals could also provide actionable guidance for fine-tuning and calibration, with the potential to guide models toward more reliable reasoning trajectories. LT-derived metrics could be incorporated as training-time regularizers that encourage more aligned and productive trajectories. In addition, our study introduced lightweight techniques for metric aggregation and threshold selection. An exciting direction for future work is to explore learned classifiers or ensembles to further boost the informativeness of the signals. Finally, LT signals are designed as predictive, representation-level indicators of model performance, not as direct causal explanations. Uncovering the causal mechanisms underlying these trajectories remains an open and compelling question for future interpretability research.

REPRODUCIBILITY STATEMENT

Section 3 outlines the models, datasets, and methodology used in detail. Section K in Appendix describes inference parameters in detail. The data and code can be found at https://github.com/martinagvilas/LT-signals.

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

# A LATENT-TRAJECTORY SIGNALS

To investigate how LT dynamics relate to model performance, we compare distributions of the three signals—Net Change, Cumulative Change, and Aligned Change—conditioned on whether a model's final answer was correct or incorrect. Figures 7, 8 and 9 present box plots of these metrics across datasets and model families. These visualizations allow us to assess whether systematic differences in LT signals are associated with answer correctness, and whether such effects are consistent across evaluation settings.

Across all three signals, consistent patterns emerge. Net Change values (Figure 7) are generally higher for correct responses than for incorrect responses, suggesting that successful reasoning is associated with overall larger representational drifts. In contrast, Cumulative Change values (Figure 8) are often larger for incorrect responses, indicating that when models answer incorrectly, their latent trajectories tend to involve more movement through representational space, potentially reflecting less stable reasoning. Finally, Aligned Change values (Figure 9) are again higher for correct responses, implying that effective reasoning requires updates that advance more directly towards the final state.

Taken together, these results suggest that correct predictions are characterized by a larger overall representational shift, accompanied by trajectories that are more directionally consistent, whereas incorrect predictions tend to involve longer, less aligned paths through latent space, reflecting noisier and less stable reasoning trajectories. This pattern holds across models and datasets, indicating that LT signals provide complementary and reliable signals of reasoning quality.

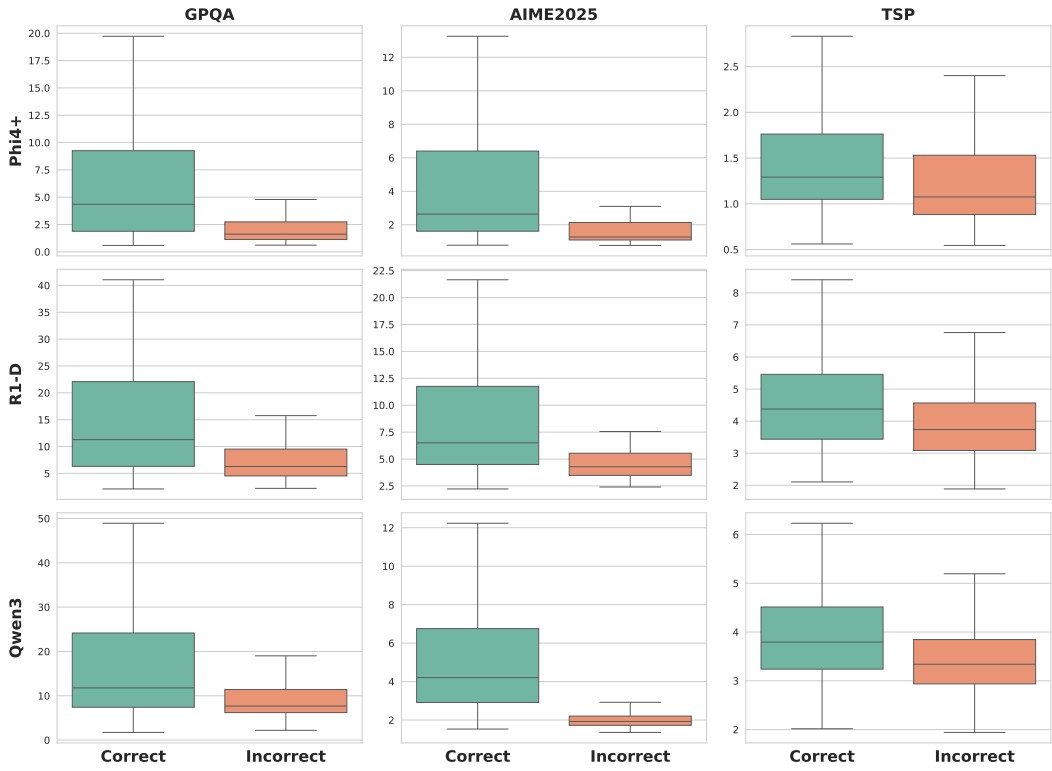

Figure 7: Distribution of Net Change by accuracy. Values are generally higher for correct than for incorrect responses, suggesting that successful reasoning is associated with overall larger representational drifts, which may be a sign of deeper reasoning.

Figure 8: Distribution of Cumulative Change by accuracy. Values are often larger for incorrect responses, indicating that when models answer incorrectly, their latent trajectories tend to involve more movement through representational space, potentially reflecting less stable reasoning.

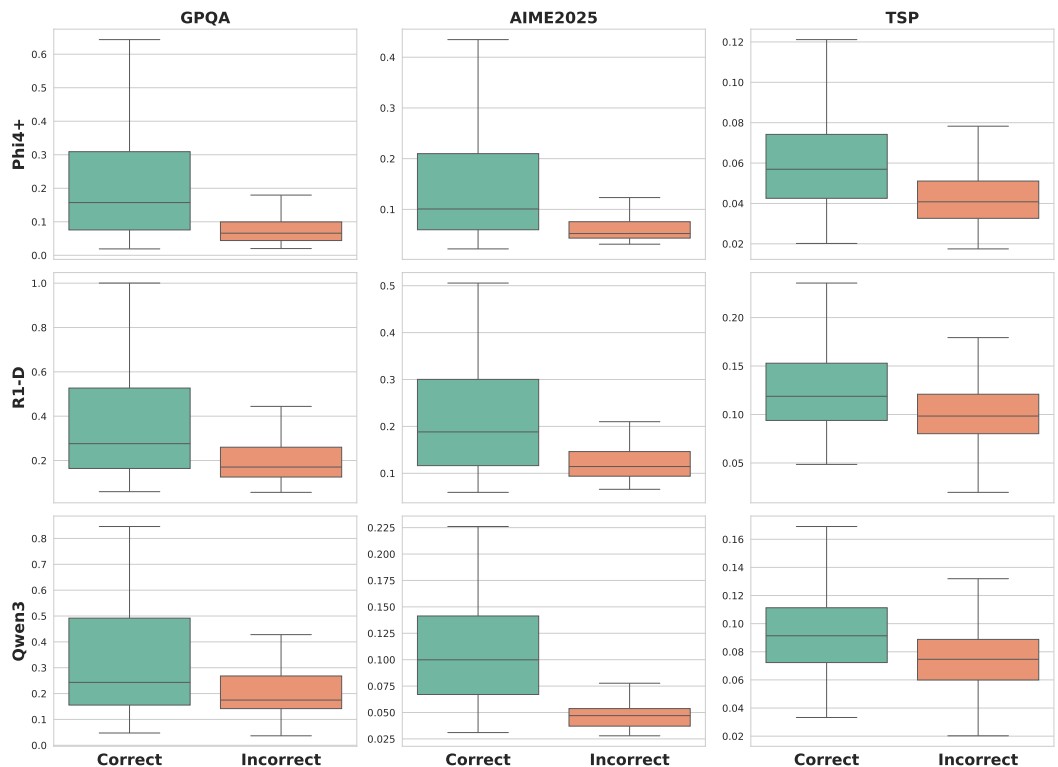

Figure 9: Distribution of Aligned Change by accuracy. Values are higher for correct responses, implying that effective reasoning involves intermediate representational updates that advance more directly towards the final state.

We also provide the average layer-wise values of each LT signal in Figure 10, 11, and 12. Each subplot compares how internal changes evolve by layer.

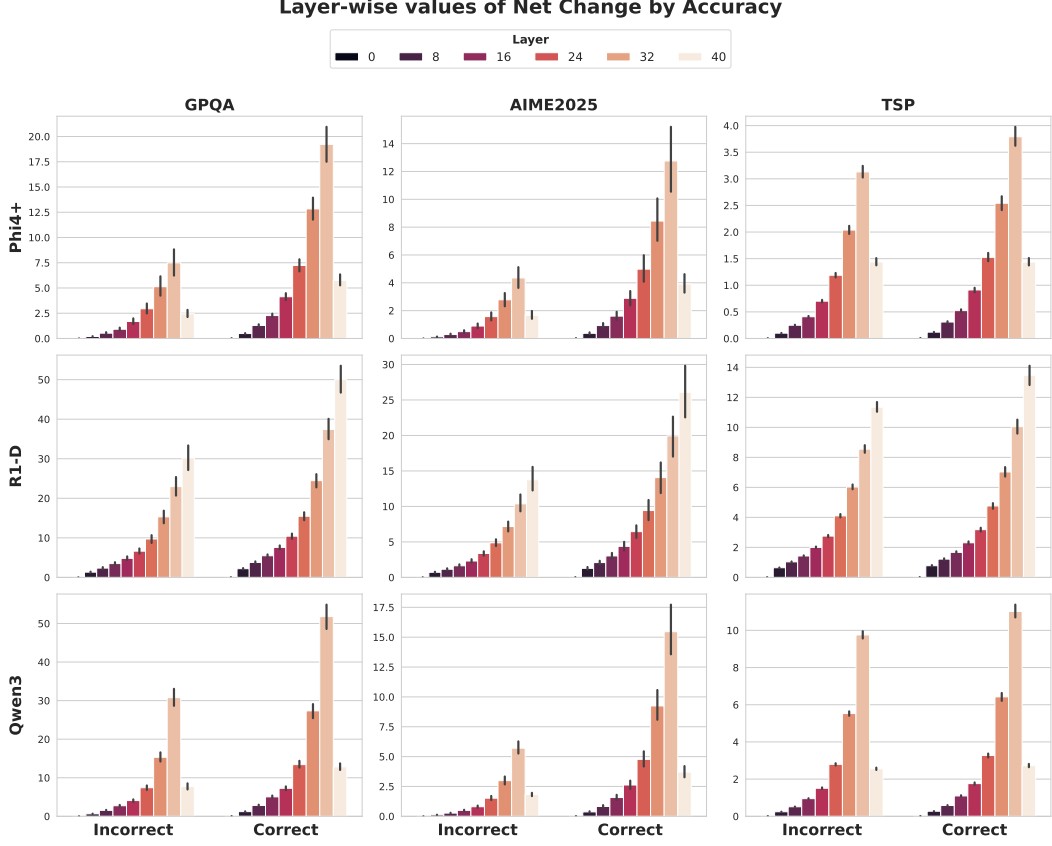

Figure 10: Layer-wise Net Change values for correct vs. incorrect reasoning traces across models and benchmarks. Correct trajectories generally show larger representational shifts across layers compared to incorrect ones, indicating that stronger changes are associated with successful reasoning.

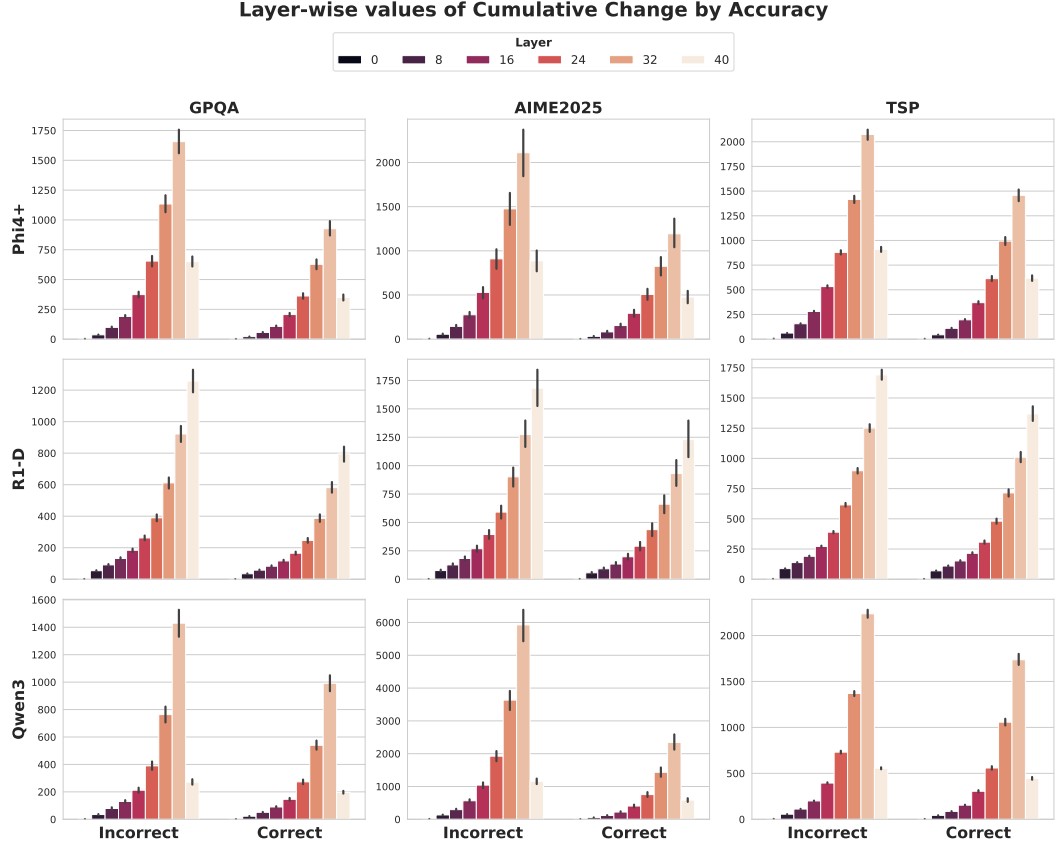

Figure 11: Layer-wise Cumulative Change values for correct vs. incorrect reasoning traces across models and benchmarks. Incorrect trajectories accumulate substantially larger representational shifts across layers than correct ones, indicating that correct traces take more direct paths towards the final solution.

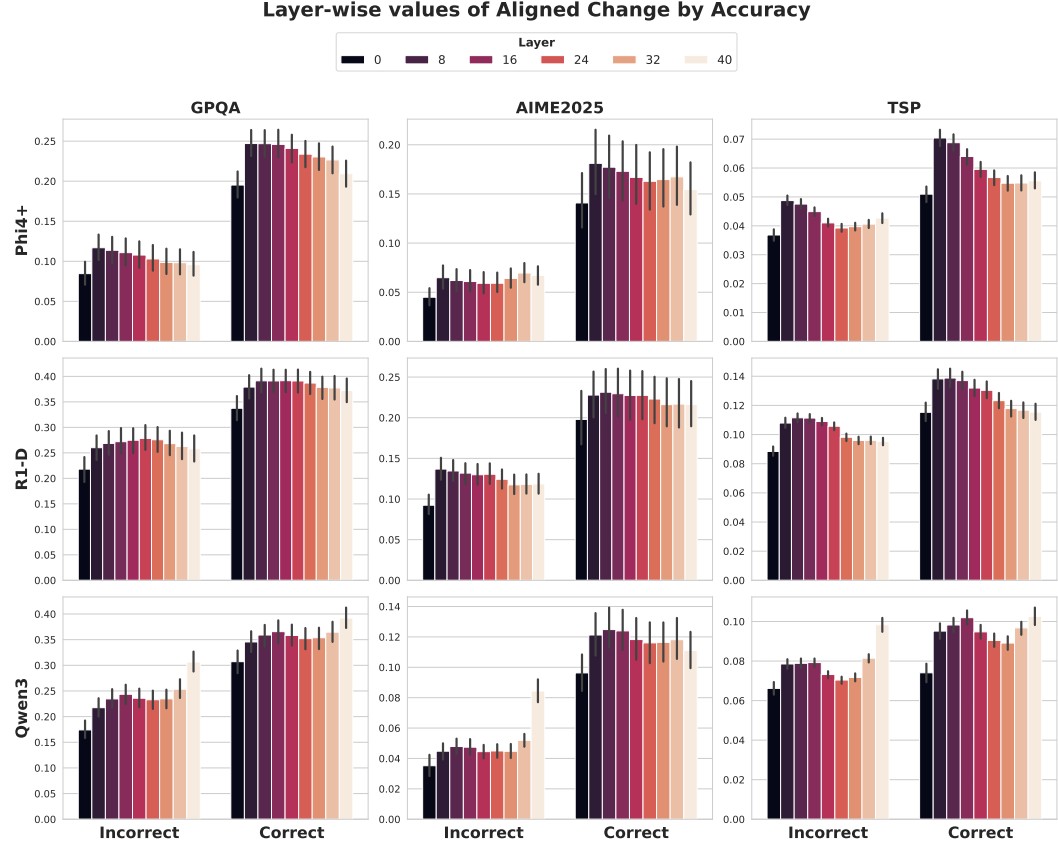

Figure 12: Layer-wise Aligned Change values for correct vs. incorrect reasoning traces across models and benchmarks. Correct trajectories consistently exhibit stronger alignment across layers compared to incorrect ones.

# B  PREDICTIVITY OF LATENT-TRAJECTORY SIGNALS

In Table 3 we provide the ROC-AUC and correlation (Spearman's $r$) values with accuracy for each Latent-Trajectory (LT) and baseline metric computed per model-dataset pair.

Table 3: ROC-AUC and correlation (Spearman's $r$) with accuracy for each model-dataset pair. Cumulative Change and Layer Angle were signed reversed for ROC-AUC computation.

| Model | Metric | GPQA AUC | GPQA Corr. | AIME2025 AUC | AIME2025 Corr. | TSP AUC | TSP Corr. |
|---|---|---|---|---|---|---|---|
| R1-D | Net Change | .688 | .320 | .757 | .433 | .641 | .223 |
| | Cumulative Change | .690 | -.323 | .697 | -.333 | .687 | -.294 |
| | Aligned Change | .670 | .288 | .755 | .430 | .662 | .255 |
| | Layer Magnitude | .606 | .180 | .795 | .497 | .449 | -.080 |
| | Layer Angle | .392 | .184 | .685 | -.313 | .740 | -.378 |
| | Logit Margin | .554 | .092 | .597 | .163 | .656 | .246 |
| | Entropy | .510 | .016 | .488 | -.020 | .588 | .138 |
| | Perplexity | .656 | .266 | .558 | .097 | .597 | .153 |
| PhiR+ | Net Change | .744 | .391 | .755 | .366 | .625 | .433 |
| | Cumulative Change | .740 | -.384 | .786 | -.410 | .785 | -.333 |
| | Aligned Change | .744 | .391 | .755 | .366 | .733 | .430 |
| | Layer Magnitude | .730 | .368 | .625 | .180 | .768 | .497 |
| | Layer Angle | .663 | -.260 | .727 | -.325 | .733 | -.312 |
| | Logit Margin | .652 | .243 | .535 | .050 | .438 | .163 |
| | Entropy | .230 | -.321 | .440 | -.086 | .426 | -.020 |
| | Perplexity | .369 | -.194 | .517 | .024 | .399 | .097 |
| Qwen3 | Net Change | .671 | .286 | .921 | .651 | .637 | .229 |
| | Cumulative Change | .655 | -.259 | .947 | -.691 | .748 | -.414 |
| | Aligned Change | .635 | .225 | .909 | .632 | .679 | .300 |
| | Layer Magnitude | .557 | .095 | .295 | -.317 | .387 | -.189 |
| | Layer Angle | .509 | -.015 | .878 | -.584 | .727 | .380 |
| | Logit Margin | .444 | -.094 | .728 | .351 | .602 | .170 |
| | Entropy | .513 | .022 | .286 | -.324 | .371 | -.213 |
| | Perplexity | .272 | -.380 | .454 | -.071 | .591 | .151 |

Our main experiments focus on domains where previous work has shown that reasoning models exhibit stable and well-understood behavior (i.e., scientific, mathematical, and algorithmic tasks), and thus where these models are mainly applied to. To further assess whether our latent-trajectory signals also generalize to more "divergent" forms of reasoning, we expanded our analysis to include tasks that require more commonsense-based, creative, narrative, or socially grounded forms of reasoning.

Specifically, we evaluated our metrics on two BIG-Bench tasks (Srivastava et al., 2023) where existing reasoning models have not saturated performance and where the required cognitive processes differ substantively from the domains tested in the main paper: (i) *Understanding Fables*: Measures the ability of models to identify the most suitable moral for a given fable, targeting narrative abstraction, commonsense, and creativity; (ii) *Social IQa*: Measures the ability of models to reason about the common-sense implications of social situations, targeting social reasoning, emotional inference, and context-sensitive commonsense. Because the new tasks generate shorter reasoning paths, we report the results obtained when averaging reasoning segments of 100 tokens.

In addition to extending our analysis to new reasoning domains, we also verify that our metrics are predictive in open-source models of different sizes and architectural family. Concretely, we probe DeepSeek-R1-Llama 8B, DeepSeek-R1-Qwen 32B, and DeepSeek-R1-Llama 70B.

As Table 4 shows, we observe that our LT signals remain significantly predictive of answer correctness across the new models and reasoning domains. The ROC-AUC values for Net, Cumulative, and Aligned Change remain well above chance and are comparable to those reported for the main benchmarks (Net Change mean ROC-AUC $= 0.71 \pm 0.09$; Cumulative Change $= 0.72 \pm 0.09$; Aligned Change $= 0.70 \pm 0.10$). The cross-layer signals exhibit substantially higher variability across models

Table 4: ROC-AUC scores of LT metrics and Cross-Layer signals extended to models of different sizes and architectures, and reasoning problems of different domains. Reported with segment sizes of 100 tokens.

| Model | Change | GPQA | AIME25 | TSP | BB Fables | BB Social |
|---|---|---|---|---|---|---|
| R1-Qwen-14B | Net | .694 | .836 | .636 | .786 | .658 |
| R1-Qwen-14B | Cumulative | .710 | .732 | .685 | .660 | .618 |
| R1-Qwen-14B | Aligned | .709 | .720 | .570 | .674 | .623 |
| R1-Qwen-14B | Layer Mag. | .605 | .788 | .693 | .710 | .556 |
| R1-Qwen-14B | Layer Ang. | .388 | .672 | .387 | .354 | .457 |
| Phi4-R+-14B | Net | .708 | .807 | .708 | .778 | .713 |
| Phi4-R+-14B | Cumulative | .750 | .802 | .786 | .775 | .714 |
| Phi4-R+-14B | Aligned | .752 | .797 | .718 | .777 | .714 |
| Phi4-R+-14B | Layer Mag. | .773 | .773 | .780 | .773 | .743 |
| Phi4-R+-14B | Layer Ang. | .808 | .798 | .814 | .799 | .780 |
| Qwen3-14B | Net | .665 | .937 | .668 | .839 | .696 |
| Qwen3-14B | Cumulative | .687 | .947 | .713 | .844 | .683 |
| Qwen3-14B | Aligned | .689 | .910 | .531 | .845 | .690 |
| Qwen3-14B | Layer Mag. | .377 | .787 | .696 | .760 | .559 |
| Qwen3-14B | Layer Ang. | .806 | .806 | .707 | .818 | .656 |
| R1-Llama-8B | Net | .633 | .888 | .514 | .653 | .595 |
| R1-Llama-8B | Cumulative | .628 | .894 | .582 | .651 | .587 |
| R1-Llama-8B | Aligned | .642 | .900 | .525 | .657 | .590 |
| R1-Llama-8B | Layer Mag. | .399 | .784 | .623 | .618 | .572 |
| R1-Llama-8B | Layer Ang. | .448 | .865 | .448 | .471 | .464 |
| R1-Qwen-32B | Net | .705 | .805 | .662 | .756 | .639 |
| R1-Qwen-32B | Cumulative | .691 | .814 | .690 | .732 | .629 |
| R1-Qwen-32B | Aligned | .712 | .821 | .579 | .746 | .639 |
| R1-Qwen-32B | Layer Mag. | .622 | .403 | .667 | .610 | .590 |
| R1-Qwen-32B | Layer Ang. | .671 | .815 | .606 | .699 | .635 |
| R1-Llama-70B | Net | .694 | .837 | .696 | .786 | .658 |
| R1-Llama-70B | Cumulative | .697 | .856 | .707 | .804 | .652 |
| R1-Llama-70B | Aligned | .695 | .857 | .701 | .805 | .655 |
| R1-Llama-70B | Layer Mag. | .417 | .169 | .620 | .510 | .516 |
| R1-Llama-70B | Layer Ang. | .326 | .705 | .595 | .492 | .497 |

and datasets (Cross-Layer Magnitude Change mean ROC-AUC $= 0.56 \pm 0.18$; Cross-Layer Angle Change $= 0.56 \pm 0.18$).

These results suggest that the temporal structure we capture in latent space reflects a broader phenomenon in reasoning-oriented model behavior, not one limited to particular architectures or problem-solving tasks. We believe these metrics remain predictive even in more "creative" tasks because they quantify relative differences between correct and incorrect trajectories rather than assuming any particular structure of the task. Even if a domain naturally produces larger or more exploratory latent movements overall, correct trajectories still tend to show more consistent and directed progression toward their final internal state, whereas incorrect ones typically wander more or exhibit less alignment. Thus, the absolute scale of representational change may vary across domains, but the contrast between successful and unsuccessful paths remains robust, which is what the LT signals capture.

## C  CALIBRATION PROCEDURE FOR THRESHOLD SELECTION

We use a three-fold shuffled cross-validation procedure. In each split, 30% of the data is set aside for calibration and the remaining 70% is reserved for testing, with the same random seed applied across folds to ensure consistency.

**Candidate thresholds.**  Within each calibration fold and for each metric, we focus on the subset of datapoints where the model's answer was incorrect. If this subset contains fewer than 15 examples or if the metric has no valid values, we default to using the median value of the metric on the calibration set as the threshold. Otherwise, we construct a grid of candidate thresholds by taking the 20th through 99th percentiles of the metric values among the incorrect examples.

**Evaluating a candidate threshold.**  For each threshold in this grid, calibration datapoints are divided into two groups. The first group consists of datapoints where at least one candidate solution exceeds the threshold. For these, we accept the first candidate that crosses the threshold and record its accuracy. The second group contains the remaining datapoints, which are resolved using majority vote across their candidates. The overall calibration accuracy for a given threshold is computed as the weighted average of the accuracies from these two groups, proportional to their sizes.

**Selecting the threshold.**  We then rank thresholds by their overall calibration accuracy. The two best-performing thresholds are identified, and we set the final calibration threshold to their median.

**Direction of comparison.**  For most metrics, higher values indicate stronger signals, so the threshold rule is applied as $\text{metric} \geq t$. The Cumulative Change signals behave in the opposite direction, with smaller values being more predictive; here the rule is applied as $\text{metric} \leq t$.

# D   COMBINED LATENT-TRAJECTORY SCORE

In addition to exploring each metric separately, we built a **Combined LT score**. For each dataset, we quantified the predictive utility of each signal by calculating its absolute Pearson correlation with accuracy on a 10% calibration slice of the dataset. We align directions so that larger values always indicate better performance (i.e. Cumulative Change was sign-inverted). We then normalize the correlations to obtain weights that sum to one, which yields an interpretable distribution of relative importance across metrics. The combined LT score for each sampled solution is a weighted sum of the LT values, where signals that are more strongly associated with accuracy on the calibration set contributed more to the final score. Table 5 reports the weights obtained.

Table 5: Metric weights for Combined Latent Space score.

| Model | Dataset | Net Change | Cumulative Change | Aligned Change |
|---|---|---|---|---|
| R1-D | GPQA | 0.35 | 0.40 | 0.25 |
| PhiR+ | GPQA | 0.30 | 0.38 | 0.31 |
| Qwen3 | GPQA | 0.43 | 0.25 | 0.32 |
| R1-D | AIME2025 | 0.31 | 0.35 | 0.34 |
| PhiR+ | AIME2025 | 0.26 | 0.45 | 0.29 |
| Qwen3 | AIME2025 | 0.30 | 0.43 | 0.28 |
| R1-D | TSP | 0.24 | 0.39 | 0.37 |
| PhiR+ | TSP | 0.20 | 0.38 | 0.42 |
| Qwen3 | TSP | 0.19 | 0.43 | 0.37 |

# E    LATENT-TRAJECTORY SIGNALS FOR INFERENCE-TIME SCALING

In the main text, we evaluated efficiency and reliability when applying LT thresholds in a sequential inference procedure. Here, we provide additional analyses focusing on the subset of samples that exceeded the thresholds. This allows us to directly quantify (i) the accuracy of solutions accepted early and (ii) the proportion of datapoints where the LT decision rule was triggered.

Table 6 reports accuracy and coverage for above-threshold samples across models and datasets. Accuracy here refers only to the subset of candidate solutions whose LT score surpassed the calibrated threshold, while the "Datapoints" column indicates the fraction of evaluation datapoints where an early stop occurred. As expected, above-threshold samples are consistently more accurate than the overall average, often approaching ceiling performance for stricter thresholds. At the same time, the coverage varies: some learned thresholds are more lenient, allowing the rule to apply to a larger fraction of datapoints, while others are stricter, isolating a smaller but more reliable subset.

Table 6: Above threshold evaluation across models and datasets with LT (LT) strategies. Accuracy (%) of samples above threshold, and percentage of Datapoints where LT decision rule was applied.

| Model | Strategy | GPQA | | AIME2025 | | TSP | |
| | | Acc. (%) | Datapoints (%) | Acc. (%) | Datapoints (%) | Acc. (%) | Datapoints (%) |
| --- | --- | --- | --- | --- | --- | --- | --- |
| R1-D | LT – Net | 64.60 | 88.2 | 68.73 | 90.5 | 28.73 | 99.4 |
| | LT – Cumulative | 67.57 | 81.4 | 75.63 | 54.0 | 32.13 | 96.4 |
| | LT – Aligned | 63.77 | 92.0 | 73.00 | 82.5 | 31.60 | 93.4 |
| | LT – Combined | 66.50 | 81.2 | 78.53 | 65.1 | 30.53 | 98.5 |
| PhiR+ | LT – Net | 84.37 | 60.0 | 82.20 | 80.9 | 42.03 | 97.0 |
| | LT – Cumulative | 84.87 | 57.1 | 89.53 | 74.6 | 56.20 | 74.7 |
| | LT – Aligned | 88.10 | 44.6 | 89.33 | 71.4 | 49.73 | 88.7 |
| | LT – Combined | 86.30 | 48.0 | 91.10 | 69.8 | 51.53 | 81.5 |
| Qwen3 | LT – Net | 65.17 | 95.6 | 87.13 | 92.1 | 45.87 | 61.6 |
| | LT – Cumulative | 69.47 | 74.1 | 96.50 | 85.7 | 37.90 | 95.8 |
| | LT – Aligned | 64.73 | 87.7 | 85.37 | 95.2 | 40.50 | 87.5 |
| | LT – Combined | 65.53 | 88.2 | 85.37 | 95.2 | 47.43 | 74.4 |

To further illustrate this tradeoff, Figure 13 plots accuracy as a function of threshold quantiles. Higher quantiles consistently yield higher accuracy across all metrics, models, and datasets, indicating that LT signals reliably concentrate correct solutions in their upper ranges. In several cases, accuracy at the top quantiles approaches $100\%$, demonstrating that filtering by strong LT signals isolates highly reliable reasoning traces.

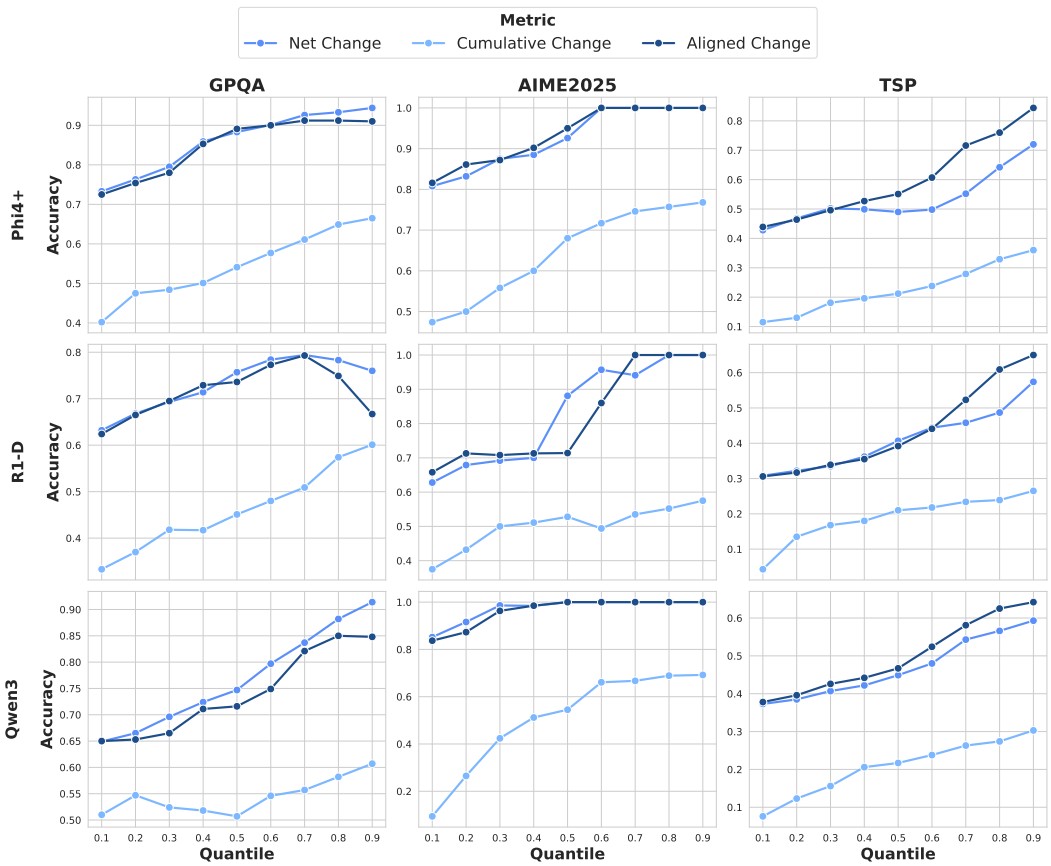

Figure 13: Accuracy of datapoints above thresholds defined over a range of quantiles. Given that Cumulative Change is negatively correlated with accuracy, we inverted the quantile selection (i.e., replacing $q$ with $1 - q$) so that higher quantiles consistently correspond to higher expected accuracy. For all metrics, accuracy increases consistently with higher quantiles across datasets and models. This evidences that these metrics are predictive of answer quality.

# F   COMPARISON AGAINST FIRST SAMPLE BASELINE

To further evaluate the robustness of our mutli-sample selection strategy, we compare against a simpler baseline where the first sample of each question is kept as the final choice.

As shown in Table 7, this heuristic is occasionally competitive, but it is also unreliable: while it matches LT performance in a few cases and is slightly higher in a small number of settings, it also suffers substantial drops in others (e.g., $-6$ to $-10$ points on some complex datasets). This variability confirms that outcome quality for early samples is highly inconsistent across tasks and models, meaning there is no dependable way to know when the first or second sample will be much worse. In contrast, LT-based selection offers a principled and stable alternative. It uses information already produced during decoding (so computational cost is essentially unchanged), and it delivers consistently strong performance, often yielding notable gains on more challenging datasets. Even in cases where LT is slightly below the first-sample heuristic, the differences are small, whereas the improvements in the opposite direction can be large.

Table 7: Accuracy with Latent-Trajectory (LT) signals. All accuracies are expressed in percent. "First Sample" uses first-sample baseline accuracies. $\Delta$ is the accuracy gain relative to this baseline.

| Model | Strategy | GPQA (Acc / $\Delta$) | AIME2025 (Acc / $\Delta$) | TSP (Acc / $\Delta$) |
|---|---|---|---|---|
| R1-D | First Sample | 60.41 | 56.67 | 28.13 |
| | LT–Net | 61.10 (+0.69) | 61.90 (+5.23) | 28.60 (+0.47) |
| | LT–Cumulative | 62.10 (+1.69) | 58.70 (+2.03) | 30.90 (+2.77) |
| | LT–Aligned | 61.10 (+0.69) | 60.30 (+3.63) | 29.50 (+1.37) |
| Phi4R+ | First Sample | 69.70 | 73.33 | 43.13 |
| | LT–Net | 68.80 (-0.90) | 79.40 (+6.07) | 42.30 (-0.83) |
| | LT–Cumulative | 69.60 (-0.10) | 81.00 (+7.67) | 44.40 (+1.27) |
| | LT–Aligned | 69.60 (-0.10) | 82.50 (+9.17) | 44.10 (+0.97) |
| Qwen3 | First Sample | 63.13 | 76.67 | 36.25 |
| | LT–Net | 63.70 (+0.57) | 79.40 (+2.73) | 35.40 (-0.85) |
| | LT–Cumulative | 63.30 (+0.17) | 84.10 (+7.43) | 36.30 (+0.05) |
| | LT–Aligned | 64.20 (+1.07) | 80.90 (+4.23) | 37.80 (+1.55) |

# G    Cross-Dataset Generalization in Sequential Strategies

To assess whether LT thresholding requires dataset-specific tuning or can generalize across reasoning tasks, we conduct a cross-dataset threshold transfer experiment. Our thresholding mechanism is percentile-based: cross-validation selects the optimal percentile of the combined LT score distribution rather than a fixed numeric value. This percentile is then mapped to each dataset's distribution at evaluation time.

For each model, we perform the following procedure: (i) Use cross-validation on one dataset (e.g., GPQA) to select the optimal Combined LT score percentile; (ii) Apply the same percentile, without re-tuning, to all other datasets. Each test dataset computes the corresponding numeric threshold using its own metric distribution; (iii) Repeat for all train-test dataset pairs across GPQA, AIME2025, and TSP. Reported values in the table are averaged over the test datasets for each model.

The results (see Table 8) show that a single global percentile per model generalizes well: accuracy remains close to MV@5 or improves, while the LT–Combined strategy consistently reduces sample usage by 20–70%. This demonstrates that LT-based thresholding does not rely on dataset-specific calibration and maintains strong performance when transferred across heterogeneous reasoning domains.

Table 8: Cross-dataset threshold generalizability.

| Model | Strategy | GPQA | | AIME2025 | | TSP | |
|---|---|---|---|---|---|---|---|
| | | Acc. (avg % / ΔAcc) | Samples (avg / ΔTok %) | Acc. (avg % / ΔAcc) | Samples (avg / ΔTok %) | Acc. (avg % / ΔAcc) | Samples (avg / ΔTok %) |
| R1-D | MV@5 | 59.90 | 5.00 | 56.67 | 5.00 | 27.50 | 5.00 |
| | LT – Combined | 60.40 (+0.5) | 2.11 (+41.5) | 60.00 (+3.3) | 1.40 (+60.3) | 28.45 (+1.0) | 1.59 (+67.3) |
| Phi4R+ | MV@5 | 70.20 | 5.00 | 80.00 | 5.00 | 41.25 | 5.00 |
| | LT – Combined | 69.20 (-1.0) | 2.64 (+25.5) | 80.00 (+0.0) | 2.87 (+23.1) | 41.85 (+0.6) | 2.52 (+45.2) |
| Qwen3 | MV@5 | 63.96 | 5.00 | 70.00 | 5.00 | 36.25 | 5.00 |
| | LT – Combined | 63.60 (-0.4) | 2.08 (+45.4) | 76.65 (+6.7) | 2.18 (+40.5) | 35.90 (-0.3) | 1.29 (+73.6) |

## H CROSS-DATASET GENERALIZATION IN EARLY EXIT STRATEGIES

To evaluate whether early in the trace exit experiments can be generalized across reasoning tasks, we train a random forest classifier from multiple datasets and evaluate it using a leave-one-dataset-out protocol. Concretely, for each model, we pool the LT signals from two datasets and train a Random Forest classifier on this combined set, using only the early-step features. The third dataset is held out entirely and used solely for evaluation. At test time, the trained Random Forest is applied to the held-out dataset to score partial chains of thought and select the predicted best reasoning path. This design directly measures cross-dataset transfer: the selector must generalize to a reasoning domain it has never seen, demonstrating that early-trace selection does not require dataset-specific tuning.

Table 9 reports the averaged values of the cross-dataset classifier accuracies on the first 4 reasoning segments (composed of 500 tokens). Results show that LT-based early trace selection generalizes well across datasets: a classifier trained on two datasets maintains competitive accuracy on the held-out dataset while saving 60–75% of tokens. Despite having no access to task-specific labels, LT matches or exceeds the Maj@5 baseline in several settings, demonstrating that early-trace signals are transferable across heterogeneous reasoning tasks. Accuracy differences remain small (typically within ±2%), confirming that strong cross-dataset performance can be achieved without dataset-specific tuning.

Table 9: Cross-Dataset generalizability early in the trace.

| Model | Strategy | GPQA Accuracy (% / Δ%) | Saved Tokens (Δ%) | AIME2025 Accuracy (% / Δ%) | Saved Tokens (Δ%) | TSP Accuracy (% / Δ%) | Saved Tokens (Δ%) |
|---|---|---|---|---|---|---|---|
| R1-D | Maj@5 | 59.90 | - | 56.67 | - | 27.50 | - |
|  | LT | 60.91 (+1.0) | +59.7 | 56.67 (+0.0) | +59.2 | 30.78 (+3.3) | +69.4 |
| PhiR+ | Maj@5 | 70.20 | - | 80.00 | - | 41.25 | - |
|  | LT | 69.07 (-1.1) | +70.4 | 82.50 (+2.5) | +71.3 | 40.00 (-1.3) | +75.1 |
| Qwen3 | Maj@5 | 63.96 | - | 70.00 | - | 36.25 | - |
|  | LT | 62.31 (-1.7) | +60.7 | 73.33 (+3.3) | +73.4 | 37.03 (+0.8) | +71.1 |

# I ACCURACY ACROSS EARLY STEPS

To examine how early checkpoint selection affects the robustness of our metric, we evaluate the accuracy of the LT-based approach at checkpoints taken every 500 tokens over the first 2500 tokens of each trace.

As shown in Figure 14, LT-based accuracy remains both significant and stable across early checkpoint choices. The average variation across segments is 0.05.

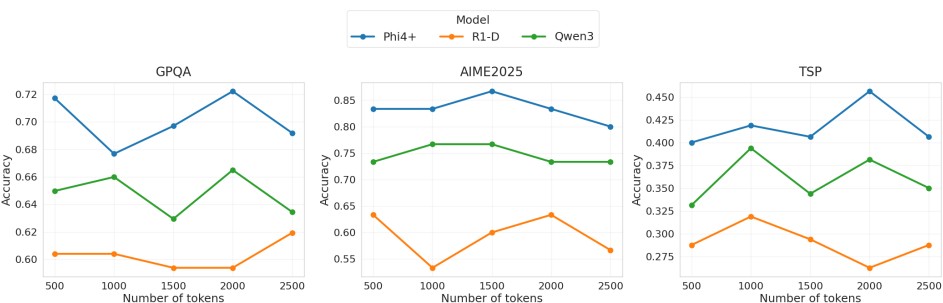

Figure 14: Latent-Trajectory-based accuracies across early reasoning segments.

## J    REPRESENTATIONAL AVERAGING

To enhance the signal robustness and reduce the dimensionality of the reasoning trace, we partition it into non-overlapping reasoning segments of size 500. For each layer, we then average the token representations within each segment. We found that this procedure preserves the overall trajectory of the reasoning process.

The choice of 500 tokens was guided by the average answer lengths across datasets. The dataset with the shortest responses still had an average of 5k tokens per answer. Setting the window to 500 tokens therefore ensures that, on average, we obtain at least 10 measurement points per answer in this dataset, and proportionally more in the others.

To further demonstrate that LT signals can still be predictive of shorter reasoning traces, Table 10 demonstrates how our ROC-AUC results are equivalent when considering interval segments of other sizes. For each LT signal, the fluctuations across segment lengths are overall small (with an average variation across segment lengths of $0.01 - 0.05$ for a model-dataset pair).

| Model | Dataset | Net Change | | | | Cumulative Change | | | | Aligned Change | | | |
|---|---|---|---|---|---|---|---|---|---|---|---|---|---|
| | | 100 | 300 | 500 | 700 | 100 | 300 | 500 | 700 | 100 | 300 | 500 | 700 |
| R1-D | GPQA | 0.69 | 0.69 | 0.69 | 0.68 | 0.71 | 0.70 | 0.69 | 0.68 | 0.71 | 0.68 | 0.67 | 0.66 |
| | AIME2025 | 0.70 | 0.74 | 0.76 | 0.76 | 0.73 | 0.72 | 0.70 | 0.71 | 0.72 | 0.75 | 0.76 | 0.74 |
| | TSP | 0.64 | 0.61 | 0.64 | 0.66 | 0.69 | 0.69 | 0.69 | 0.69 | 0.57 | 0.64 | 0.66 | 0.68 |
| Phi4R+ | GPQA | 0.75 | 0.74 | 0.74 | 0.75 | 0.75 | 0.74 | 0.74 | 0.74 | 0.75 | 0.74 | 0.74 | 0.73 |
| | AIME2025 | 0.78 | 0.77 | 0.75 | 0.79 | 0.80 | 0.79 | 0.79 | 0.78 | 0.80 | 0.77 | 0.76 | 0.77 |
| | TSP | 0.71 | 0.67 | 0.62 | 0.71 | 0.79 | 0.78 | 0.78 | 0.78 | 0.72 | 0.68 | 0.73 | 0.75 |
| Qwen3 | GPQA | 0.67 | 0.67 | 0.67 | 0.65 | 0.69 | 0.67 | 0.66 | 0.65 | 0.69 | 0.66 | 0.63 | 0.62 |
| | AIME2025 | 0.94 | 0.93 | 0.92 | 0.92 | 0.95 | 0.95 | 0.95 | 0.95 | 0.91 | 0.93 | 0.91 | 0.92 |
| | TSP | 0.67 | 0.63 | 0.64 | 0.63 | 0.71 | 0.74 | 0.75 | 0.76 | 0.53 | 0.65 | 0.68 | 0.70 |

Table 10: AUC–ROC results across segment lengths

In addition, we compared fixed-$k$ strategy to defining segments by newline tokens, as other studies report (Sun et al., 2025). However, segment sizes varied substantially across models under this approach, making it less comparable across architectures.

## K   MODELS AND INFERENCE SETTINGS

We perform our inference and evaluation using the **Eureka ML Insights** framework (`microsoft/eureka-ml-insights`).

We used a max generation length of 31,768 tokens for all models. For all experiments, we report model sources and inference parameters to ensure reproducibility:

**DeepSeek-R1-Qwen 14B**   `deepseek-ai/DeepSeek-R1-Distill-Qwen-14B`

- Temperature = 0.6
- Top-$p$ = 0.95

**Phi-4-Reasoning-Plus**   `microsoft/Phi-4-reasoning-plus`

- Temperature = 0.8
- Top-$k$ = 50
- Top-$p$ = 0.95

**Qwen3 14B (thinking enabled)**   `Qwen/Qwen3-14B`

- Temperature = 0.6
- Top-$p$ = 0.95
- Top-$k$ = 20

**DeepSeek-R1-Llama 8B**   `deepseek-ai/DeepSeek-R1-Distill-Llama-8B`

- Temperature = 0.6
- Top-$p$ = 0.95

**DeepSeek-R1-Qwen 32B**   `deepseek-ai/DeepSeek-R1-Distill-Qwen-32B`

- Temperature = 0.6
- Top-$p$ = 0.95

**DeepSeek-R1-Llama 70B**   `deepseek-ai/DeepSeek-R1-Distill-Llama-70B`

- Temperature = 0.6
- Top-$p$ = 0.95

