# OpenReview forum: "Tracing the Traces: Latent Temporal Signals for Efficient and Accurate Reasoning"
_ICLR.cc/2026/Conference — ICLR 2026 Poster_

### Official Review · Reviewer_NXfb · 2025-10-31

**Soundness:** 2
**Presentation:** 3
**Contribution:** 1
**Rating:** 2
**Confidence:** 4

**Summary:**

In this work, the paper explores whether through intermediate activations we can identify chains of thought that are unlikely to lead to a correct answer and, if so, leverage these signals to guide the selection of more promising reasoning paths. They explore a number of signals, namely Net Change (measuring the overall magnitude of representational drift from initial to final hidden states), Cumulative Change (quantifying the total accumulated movement across intermediate updates), and Aligned Change (assessing how well sequential updates progress toward the final state via cosine similarity), and first investigate their correlation with accuracy. They find the signals can predict solution correctness across various science, math, and algorithmic benchmarks on several reasoning models, consistently outperforming baselines like cross-layer signals and output-distribution metrics. They use this to improve performance by guiding multi-sample inference through early answer selection and trace pruning, substantially reducing token usage while preserving or boosting accuracy on average over majority voting. The signals emerge early enough to allocate compute dynamically to promising paths.

**Strengths:**

1. **Training-Free Methodology**: The paper introduces Latent-Trajectory (LT) signals (Net Change, Cumulative Change, and Aligned Change) that leverage hidden states during reasoning without requiring additional training, annotations, or external models, making it highly practical for deployment across diverse LLMs.

2. **Superior Predictive Performance**: LT signals demonstrate robust discriminative power, achieving ROC-AUC scores of 0.71–0.74 for predicting solution accuracy across multiple datasets (GPQA, AIME 2025, TSP) and models (DeepSeek-R1-Distill-Qwen-14B, Phi4-Reasoning-Plus, Qwen3-14B), consistently outperforming baselines like cross-layer signals and output-distribution metrics.

3. **Efficiency Gains in Inference Scaling**: By enabling early answer selection and trace pruning, LT signals reduce token usage by up to 70% in multi-sample inference, addressing key challenges in compute-intensive reasoning tasks like overthinking.

**Weaknesses:**

1.  **Heuristic Approach with Unproven Scalability**: The proposed Latent-Trajectory signals are fundamentally heuristic, derived from correlations observed in a narrow set of models. The evaluation is restricted to three models of a similar size from only two architectural families (Qwen and Phi). This lack of extensive analysis across different model types and, crucially, different model sizes (e.g., 7B, 70B, or larger) makes it unclear if these signals are a fundamental property of scaled reasoning or an emergent artifact of the specific models tested. Without this verification, there are significant questions about whether the method will "survive scaling" and generalize to future, more capable architectures.

2.  **Insufficient Discussion of Training Alternatives**: The work positions itself as a necessary inference-time intervention without adequately addressing why modern training methods, such as reinforcement learning (e.g., GRPO-style approaches), are not already solving this problem. A key question left unanswered is why models cannot be trained to inherently favor productive reasoning strategies, which would be a more robust and fundamental solution than applying a post-hoc heuristic. This omission weakens the justification for needing such an inference-time fix.

3.  **Narrow Domain Scope**: The experiments are confined to three specific and relatively structured reasoning domains: scientific multiple-choice (GPQA), contest math (AIME 2025), and algorithmic optimization (TSP). The paper does not evaluate the signals on more diverse or open-ended tasks, such as commonsense reasoning, creative problem-solving, or multi-hop question answering over text. This narrow scope limits the claim that LT signals are a general-purpose tool for assessing reasoning quality across the broad spectrum of tasks LLMs are applied to.

**Questions:**

1. **Comparison to Reinforcement Learning Training**: Why pursue this inference-time heuristic approach using Latent-Trajectory signals when reinforcement learning methods like GRPO could directly train models to favor productive reasoning paths during optimization, potentially eliminating unproductive traces without the need for handcrafted, post-training interventions that may not generalize across diverse architectures?

2. **Causal Interpretability**: While LT signals correlate with accuracy, do they reveal causal mechanisms in reasoning (e.g., via interventions on hidden states), or are they just correlated signals?

3. What evidence is there that such trace signals will generalise to other model classes and sizes and on different problems

---

> ### Author Response · Authors · 2025-11-21
> **Part 1**
>
> Thank you for your review and the feedback provided. We are grateful for your comments on the practical usefulness of a training-free approach. Below, we clarify the intended scope of the paper and address the points raised.
>
> __Diverse model classes__
>
> To address concerns about scalability, we expanded our evaluation to include models of different sizes and from a different architectural family: (1) DeepSeek-R1-Distill-Llama-8B; (2) DeepSeek-R1-Distill-Qwen-32B. These additions broaden our coverage and span the most widely used open-source reasoning architectures. While we are unable to evaluate 70B-scale models due to computational constraints, we note that three of our evaluated models are distilled to mimic the behavior from larger-scale reasoning models.
>
> As shown in the revised manuscript and in Table below, the results generalize across model sizes and architecture families, providing initial evidence that LT signals are not an artifact of a narrow configuration but reflect broader representational patterns. Across these extended settings, LT signals remain reliably predictive (average value of 0.72 and 0.09 standard deviation, reinforcing that the observed phenomena are not restricted to a narrow architecture or task type.
>
>
> __Table: ROC-AUC scores for segment size = 100__
> | **Model**               | **Change**   | **AIME 2025** | **GPQA** | **TSP** | **BB Fables** | **BB Social** |
> |-------------------------|--------------|---------------|----------|---------|----------------|---------------|
> | Deepseek-R1-Qwen14B     | Net          | 0.696         | 0.693    | 0.636   | 0.693          | 0.619         |
> | Deepseek-R1-Qwen14B     | Cumulative   | 0.732         | 0.710    | 0.685   | 0.659          | 0.618         |
> | Deepseek-R1-Qwen14B     | Aligned      | 0.720         | 0.709    | 0.570   | 0.674          | 0.623         |
> | Phi4-Reasoning-Plus     | Net          | 0.781         | 0.752    | 0.708   | 0.777          | 0.713         |
> | Phi4-Reasoning-Plus     | Cumulative   | 0.802         | 0.749    | 0.786   | 0.775          | 0.714         |
> | Phi4-Reasoning-Plus     | Aligned      | 0.797         | 0.752    | 0.718   | 0.777          | 0.714         |
> | Qwen3-14B               | Net          | 0.937         | 0.665    | 0.667   | 0.839          | 0.696         |
> | Qwen3-14B               | Cumulative   | 0.947         | 0.687    | 0.713   | 0.844          | 0.683         |
> | Qwen3-14B               | Aligned      | 0.910         | 0.689    | 0.531   | 0.845          | 0.690         |
> | Deepseek-R1-Llama8B     | Net          | 0.888         | 0.633    | 0.514   | 0.653          | 0.595         |
> | Deepseek-R1-Llama8B     | Cumulative   | 0.894         | 0.628    | 0.582   | 0.651          | 0.587         |
> | Deepseek-R1-Llama8B     | Aligned      | 0.900         | 0.642    | 0.525   | 0.657          | 0.590         |
> | Deepseek-R1-Qwen32B     | Net          | 0.805         | 0.705    | 0.662   | 0.756          | 0.639         |
> | Deepseek-R1-Qwen32B     | Cumulative   | 0.814         | 0.691    | 0.690   | 0.732          | 0.629         |
> | Deepseek-R1-Qwen32B     | Aligned      | 0.821         | 0.712    | 0.579   | 0.746          | 0.639         |
>
> Importantly, there is no theoretical reason to expect these latent-trajectory signals to behave differently across architectures or scales: they arise from high-level geometric properties of hidden-state evolution. Our empirical results, now extended to a wider size and architectural range, align with this expectation.

---

> > ### Author Response · Authors · 2025-11-21
> > **Part 2**
> >
> > __Diverse reasoning domains__
> > We appreciate this observation. We clarify that our goal is not to build a general-purpose metric for all LLM behavior, but to study latent-trajectory signals specifically in reasoning models, which are mainly trained to solve complex reasoning tasks by scaling compute at test time. These models are not primarily designed for open-ended creative generation or unverifiable tasks, and multi-hop retrieval-heavy QA is outside this scope because such tasks rely more on document retrieval and factual lookup than on extended chain-of-thought reasoning.
> >
> > That said, we agree it is important to assess whether LT signals generalize beyond the structured scientific, mathematical, and algorithmic domains in our main experiments. To directly test this, we expanded our evaluation to more diverse forms of reasoning, including commonsense, narrative abstraction, creativity, and socially grounded inference. Specifically, we added two BIG-Bench tasks where (i) current reasoning models do not saturate performance, and (ii) the cognitive demands differ substantially from the domains in the main paper:
> > - Understanding Fables (https://github.com/google/BIG-bench/tree/main/bigbench/benchmark_tasks/understanding_fables): targeting narrative abstraction, commonsense, and creativity;
> > - Social IQa (https://github.com/google/BIG-bench/tree/main/bigbench/benchmark_tasks/social_iqa): targeting social reasoning, emotional inference, and context-sensitive commonsense.
> >
> > Despite the significant shift in task style, we find that LT signals remain strongly predictive of correctness in both settings (see Table in Part 1 of our response). Net Change, Cumulative Change, and Aligned Change all achieve ROC-AUC values well above chance and comparable to those in our main benchmarks. This suggests that the temporal structure captured by LT signals reflects a broader representational phenomenon in reasoning models rather than something limited to highly convergent or quantitative problem-solving.
> >
> > We believe LT metrics remain predictive even in these more “divergent’’ reasoning tasks because they measure relative differences between successful and unsuccessful trajectories. Even when reasoning is more exploratory, correct paths tend to move more consistently toward their final representational state, while incorrect paths exhibit more wandering or inconsistent directionality. Thus, while the absolute scale of latent movement may vary by domain, the contrast between productive and unproductive trajectories remains robust.

---

> > > ### Author Response · Authors · 2025-11-21
> > > **Part 3**
> > >
> > > __Comparison to RL training__
> > >
> > > Thank you for this question, which has helped further clarify the motivation of our work.
> > > Recent RL-style reasoning optimizers (e.g., GRPO) indeed improve average reasoning quality, but they do not eliminate unproductive, divergent, or overextended reasoning traces. In fact, as mentioned in the introduction and related work section of our manuscript, several recent studies (Shojaee et al., 2025; Marjanović et al., 2025; Hassid et al., 2025) show that: (1) RL-trained reasoning models increasingly overthink, (2) they often produce long, unproductive traces, and (3) path-level variance persists even after extensive RL training. Thus, current RL methods do not inherently produce reliably structured or efficient trajectories. This limitation has motivated a line of research, cited in our introduction and related work, aimed at identifying productive reasoning paths without costly human supervision.
> > >
> > > In this regard, a key distinction is that LT signals provide instance-level assessments of trace quality, whereas RL optimizes distribution-level behavior. To our knowledge, no RL algorithm currently offers reliable per-sample correctness prediction at inference time without additional mechanisms such as verifiers, self-consistency, or post-hoc scoring. LT signals fill exactly this gap: they allow per-trace scoring with zero additional training.
> > >
> > > For this reason, LT signals are complementary to RL, not a replacement. In fact, as we note in the manuscript, LT-derived metrics could naturally be incorporated as training-time regularizers that encourage more aligned and productive trajectories. We see this integration as a promising avenue for future work, and we have clarified this more explicitly in the revised text.

---

> > > > ### Author Response · Authors · 2025-11-21
> > > > **Part 4**
> > > >
> > > > __Causal Interpretability__
> > > >
> > > > We agree that understanding causal mechanisms in reasoning models is indeed an important and related research question. However, it is outside the scope of our present work.
> > > >
> > > > Identifying causal structure in LLMs is a challenging open problem, and the difficulty is amplified in reasoning models. Classical mechanistic interpretability approaches  typically focuse on short-context, next-token mechanisms, where causal mediators can be isolated over a handful of layers or attention heads. In contrast, reasoning models operate at the opposite end of the spectrum: they involve very long chains of thought, multi-step latent computations, and extended temporal dependencies. Only a few recent papers have begun to explore whether causal interpretability methods can even extend to such long-horizon reasoning traces (e.g. [1]), and significant foundational issues remain unresolved.
> > > >
> > > > Our work therefore adopts a different but complementary interpretability perspective: representational analysis [2]. We focus on how hidden states geometrically evolve over time rather than attempting to decompose internal circuits into causal mechanisms. This approach is particularly valuable in settings where causal attribution in high-dimensional spaces is not currently feasible. Designing causal interventions for these models would require activation engineering or steering methods applied over long trajectories, which to the best of our knowledge have not yet demonstrated reliable success on reasoning models.
> > > >
> > > > We will state this explicitly in the manuscript: LT signals are designed as predictive, representation-level indicators, not as direct causal explanations. Exploring causal mechanisms underlying these trajectories is an exciting direction for future mechanistic interpretability work, but it is beyond the current capabilities of the field and beyond the scope of this paper.
> > > >
> > > > __References__
> > > >
> > > > [1] Bogdan, Paul C., et al. "Thought Anchors: Which LLM Reasoning Steps Matter?." arXiv preprint arXiv:2506.19143 (2025).
> > > >
> > > > [2] Zou, Andy, et al. "Representation engineering: A top-down approach to ai transparency." arXiv preprint arXiv:2310.01405 (2023)

---

> > ### Comment · Reviewer_NXfb · 2025-11-25
> >
> > I remain unconvinced by this statement: “Importantly, there is no theoretical reason to expect these latent-trajectory signals to behave differently across architectures or scales: they arise from high-level geometric properties of hidden-state evolution.”
> >
> > I still don’t see any theoretical justification for why these latent signals should be useful at all or why they should behave consistently across all model types and sizes. I feel the onus is on the authors to explain why these signals should work in the first place and why their properties would carry over to different architectures and scales, not the other way around.

---

> > > ### Author Response · Authors · 2025-11-28
> > > **Follow-up**
> > >
> > > Thank you for the follow-up. We appreciate your engagement and the opportunity to clarify our position on the theoretical justification for temporal latent-trajectory signals.
> > >
> > > ## Architecture-agnostic metric design by construction
> > >
> > > We would like to clarify what we mean by "no theoretical reason to expect these signals to behave differently across architectures." Our claim is not that we have proven universal applicability through first principles, but rather that the signals are defined in a way that is intentionally architecture-agnostic by construction. There is nothing in our metric definitions that "custom serves" or is tailored to any particular architecture, layer configuration, mechanism, or model family.
> > >
> > > These quantities are defined for any transformer-style model because:
> > >
> > > - **The metrics operate over token-level sequential updates**, which every autoregressive reasoning model produces. As defined in Section 3.1, for each reasoning segment n and layer l, we compute segment-level representations h̃ₗ⁽ⁿ⁾ by averaging token-level activations. From these, we derive the drift vector uₗ = h̃ₗ⁽ᴺ⁾ - h̃ₗ⁽¹⁾ and update vectors vₗ⁽ⁿ⁾ = h̃ₗ⁽ⁿ⁾ - h̃ₗ⁽ⁿ⁻¹⁾. As long as a model generates a chain-of-thought (like in any reasoning model), we can measure representational movement across its hidden activations using these basic vector operations.
> > >
> > > - **Distances and alignment metrics are purely geometric operations** on vectors in the model's residual space. Our three signals—Net Change (∥uₗ∥₂), Cumulative Change (∑∥vₗ⁽ⁿ⁾∥₂), and Aligned Change (average cosine similarity between vₗ⁽ⁿ⁾ and uₗ)—require no assumptions about internal architecture. While different architectures may have different layer mechanisms and configurations, they all produce vectors in their residual stream that can be compared using norms and cosine similarities.
> > >
> > > - **Temporal signals are more architecture-agnostic than layer-wise signals.** The baseline cross-layer signals track changes across layers within a segment, making them inherently more architecture-dependent because different models have different numbers of layers, layer functions vary across architectures, and different architectures may apply different functions at different points in the layer hierarchy. We empirically confirm this in our results: the cross-layer signals exhibit substantially higher variability across models and datasets, as shown in the table below, showing the mean and standard deviation across models and datasets:
> > >
> > > | Metric | Mean ROC-AUC | Std Deviation |
> > > |--------|--------------|---------------|
> > > | **Net Change** | 0.71 | 0.10 |
> > > | **Cumulative Change** | 0.72 | 0.09 |
> > > | **Aligned Change** | 0.70 | 0.11 |
> > > | **Cross-Layer Magnitude** | 0.56 | 0.19 |
> > > | **Cross-Layer Angle** | 0.57 | 0.21 |
> > >
> > > ## Comprehensive Empirical Validation
> > > We empirically validate this claim across substantially expanded experimental settings. Our experiments now demonstrate that these geometric signals are robustly predictive across:
> > >
> > > - **Five distinct model variants**: DeepSeek-R1-Distill-Qwen-14B, Phi-4-Reasoning-Plus, Qwen3-14B, and newly added DeepSeek-R1-Distill-Llama-8B and DeepSeek-R1-Distill-Qwen-32B (spanning three architectural families: Qwen, Phi, and Llama). We note that these three families represent the dominant paradigms in current open-source reasoning models.
> > > - **Three model scales**: 8B, 14B, and 32B parameters.
> > > - **Five diverse reasoning domains**: scientific reasoning (GPQA), mathematical problem-solving (AIME 2025), algorithmic optimization (TSP), and newly added commonsense/social reasoning (BIG-Bench Fables and Social IQa).
> > >
> > > As shown in the updated table in Part 1 of our response, LT signals achieve consistent predictive performance across all settings, with an average ROC-AUC of 0.71. The newly added domains represent substantially different reasoning demands than our original benchmarks, yet LT signals remain strongly predictive even in these settings. This directly supports the claim that the usefulness of LT signals is not an artifact of a single architecture, scale, or reasoning domain.
> > >
> > > ### Extension to 70B Scale
> > > We now additionally expand our analysis to 70B models. The table below shows results for DeepSeek-R1-Llama70B:
> > >
> > > | Model | Dataset | Metric | ROC-AUC |
> > > |-------|---------|--------|---------|
> > > | deepseek-r1-llama70B | GPQA | Net Change | 0.688 |
> > > | deepseek-r1-llama70B | GPQA | Cumulative Change | 0.681 |
> > > | deepseek-r1-llama70B | GPQA | Aligned Change | 0.672 |
> > > | deepseek-r1-llama70B | AIME2025 | Net Change | 0.851 |
> > > | deepseek-r1-llama70B | AIME2025 | Cumulative Change | 0.857 |
> > > | deepseek-r1-llama70B | AIME2025 | Aligned Change | 0.841 |
> > > | deepseek-r1-llama70B | TSP | Net Change | 0.588 |
> > > | deepseek-r1-llama70B | TSP | Cumulative Change | 0.679 |
> > > | deepseek-r1-llama70B | TSP | Aligned Change | 0.689 |
> > >
> > > These results demonstrate that LT signals maintain strong predictive performance even at 70B scale, further confirming generalization across model sizes.

---

> > > > ### Author Response · Authors · 2025-11-28
> > > > **Summary**
> > > >
> > > > In summary, we claim:
> > > > 1. The signals are **architecture-agnostic by construction**; they use only sequential hidden states and basic geometric operations, rather than tracking spatial changes across architectural layers which vary between model families.
> > > > 2. Our evaluation now spans **three major architectural families** (Qwen, Phi, Llama) at **four different scales** (8B, 14B, 32B, 70B), covering the most widely used open-source reasoning architectures. It is unclear to us what additional architectural families or model types would strengthen the generalization claim beyond this coverage.
> > > >
> > > > We believe this addresses your concern. If you have any further questions or remaining concerns, we'd be very happy to answer them. If not, we'd be very grateful if you would consider a score increase given these substantial improvements to the work.

---

### Official Review · Reviewer_SL11 · 2025-10-31

**Soundness:** 3
**Presentation:** 3
**Contribution:** 3
**Rating:** 6
**Confidence:** 3

**Summary:**

The paper proposes novel methods for evaluating the reasoning chains of language models. The chain is split into segments of length 500. For each layer, the mean hidden state of these segments is calculated. The paper suggests three measures: (1) *net change* is the L2 norm of the difference between the hidden vectors of the first and the last segment; (2) *cumulative change* is the sum of the L2 norms of the differences between consecutive hidden vectors ; and (3) *aligned change* is the average cosine similarity between each segment's update vector and the overall trend of the reasoning chain. For all measures, the mean across all layers is used to get a single scalar score. It is shown that high *net change* and high *aligned change* are associated with correct reasoning, whereas high *cumulative change* is associated with incorrect reasoning. These correlations are stronger than previously proposed measures. The paper shows how this allows for better selection between several alternative reasoning chains and even early selection of promising reasoning paths, leading to higher accuracy and efficiency.

**Strengths:**

- Clear description of the method: The paper clearly outlines the three Latent-Trajectory signals and how they are computed .
- Thorough experimental evaluation: The method is tested across multiple modern reasoning models and three distinct domains (science, math, and algorithmic problems), demonstrating robust findings .
- Immediately useful and practical results: The paper demonstrates that the proposed method can improve both the accuracy and efficiency of reasoning models. A practical advantage is that the method is training-free, it does not require costly annotations or the training of additional verifier models, making it simple to implement.

**Weaknesses:**

The proposed methods for improving performance and efficiency need some parameter tuning (e.g., threshold calibration). The segment size over which the averaged hidden states are computed is fixed at 500 (a segment size of 300 was also tested ). More fine-grained tracking of the hidden states might be insightful.

These are only minor concerns, which lead to my current vote of weak accept. If my questions below are addressed, I would consider further increasing my score.

**Questions:**

- Lines 340-353: Is the cross-validation to select the threshold done separately for each benchmark (GPQA, AIME2025, TSP), or is the same threshold used for all benchmarks? The more meaningful and fair setting in my mind would be one where a global threshold would be chosen for a model and then used for all reasoning problems.
- It makes intuitive sense that for many kinds of reasoning problems, a 'direct' or linear route through the latent space is desirable. However, other problems—particularly ones that require novel out-of-the-box solutions, lots of trial and error, divergent instead of convergent thinking —might benefit from less direct traces. How broad is the class of problems for which the presented results hold?
- More of a comment: the authors might be interested in recent work analysing the hidden states of language models using, for example, the lens of prompt entropy ("Layer by Layer: Uncovering Hidden Representations in Language Models", Skean et al., 2025) or information gain ("Measuring In-Context Computation Complexity via Hidden State Prediction", Herrmann et al., 2025).

---

> ### Author Response · Authors · 2025-11-21
> **Part 1**
>
> Thank you for the positive assessment of our work and the constructive feedback, which has helped improve our manuscript. We are grateful for your comments on the clarity of the method, the thoroughness of the experimental evaluation, and the practical usefulness of a training-free approach. Below we address each point raised in the weaknesses and question sections. We clarify experimental design decisions and present new analyses that directly address your suggestions.
>
> __Cross-dataset parameter tuning__
>
> Thank you for this helpful suggestion. Our original experiments selected the threshold separately for each model and benchmark via cross-validation. To evaluate whether a single global setting can be used across tasks, we added a new experiment where the threshold percentile is learned on one dataset and applied to all other datasets for each model.
>
> We tested all train-test cross-dataset combinations (e.g., GPQA -> AIME2025, AIME2025 -> TSP, etc.). We then averaged the results over test datasets for each model. Results (see Table below and added to the appendix of the current manuscript) show that percentile thresholds generalize well across datasets. Accuracy remains close to the within-dataset setting, and efficiency improvements (20–70% fewer samples/tokens) remain robust.
>
> This experiment confirms that LT-based selection does not require dataset-specific tuning, and a global percentile threshold can be used across heterogeneous reasoning tasks. Importantly, this also means that the method can be transferred to datasets where correct/incorrect labels are not present.
>
> **Table: Cross-dataset threshold generalizability.**
>
> | **Model** | **Strategy** | **GPQA Acc (avg % / ΔAcc)** | **GPQA Samples (avg / ΔTok%)** | **AIME2025 Acc (avg % / ΔAcc)** | **AIME2025 Samples (avg / ΔTok%)** | **TSP Acc (avg % / ΔAcc)** | **TSP Samples (avg / ΔTok%)** |
> |-----------|--------------|------------------------------|----------------------------------|-----------------------------------|--------------------------------------|------------------------------|---------------------------------|
> | **R1-D** | MV@5 | 59.90 | 5.00 | 56.67 | 5.00 | 27.50 | 5.00 |
> | | LT–Combined | 60.40 (+0.5) | 2.11 (+41.5%) | 60.00 (+3.3) | 1.40 (+60.3%) | 28.45 (+1.0) | 1.59 (+67.3%) |
> | **Phi4R+** | MV@5 | 70.20 | 5.00 | 80.00 | 5.00 | 41.25 | 5.00 |
> | | LT–Combined | 69.20 (−1.0) | 2.64 (+25.5%) | 80.00 (+0.0) | 2.87 (+23.1%) | 41.85 (+0.6) | 2.52 (+45.2%) |
> | **Qwen3** | MV@5 | 63.96 | 5.00 | 70.00 | 5.00 | 36.25 | 5.00 |
> | | LT–Combined | 63.60 (−0.4) | 2.08 (+45.4%) | 76.65 (+6.7) | 2.18 (+40.5%) | 35.90 (−0.3) | 1.29 (+73.6%) |
>
>
> In addition, we show that cross-dataset generalizability can be achieved early in the trace selection experiments. Instead of training and testing a separate classifier for each dataset and model, we follow a leave-one-out protocol where accuracy is evaluated in a held-out dataset for each model.  Concretely, for each model, we pool the LT signals from two datasets and train a RF classifier on this combined set, using only the early-step features.  The third dataset is held out entirely and used solely for evaluation.
>
> At test time, the trained Random Forest is applied to the held-out dataset to score partial chains of thought and select the predicted best reasoning path. This design directly measures cross-dataset transfer: the selector must generalize to a reasoning domain it has never seen, demonstrating that early-trace selection does not require dataset-specific tuning.
>
> Results (see Table below and now included in the appendix) show that LT-based early trace selection generalizes well across datasets: a classifier trained on two datasets maintains competitive accuracy on the held-out dataset while saving 60–75% of tokens.  LT matches or exceeds the MV baseline in several settings, demonstrating that early-trace signals are transferable across different reasoning tasks. Accuracy differences remain small (typically within ±2%), confirming that strong cross-dataset performance can be achieved without dataset-specific tuning.
>
> **Table: Cross-dataset generalization early in the trace.**
> | **Model** | **Strategy** | **GPQA Accuracy (% / Δ%)** | **GPQA Saved Tokens (Δ%)** | **AIME2025 Accuracy (% / Δ%)** | **AIME2025 Saved Tokens (Δ%)** | **TSP Accuracy (% / Δ%)** | **TSP Saved Tokens (Δ%)** |
> |-----------|--------------|-----------------------------|------------------------------|----------------------------------|----------------------------------|-----------------------------|-----------------------------|
> | **R1-D** | Maj@5 | 59.90 | – | 56.67 | – | 27.50 | – |
> | | LT | 60.91 (+1.0) | +59.7 | 56.67 (+0.0) | +59.2 | 30.78 (+3.3) | +69.4 |
> | **PhiR+** | Maj@5 | 70.20 | – | 80.00 | – | 41.25 | – |
> | | LT | 69.07 (−1.1) | +70.4 | 82.50 (+2.5) | +71.3 | 40.00 (−1.3) | +75.1 |
> | **Qwen3** | Maj@5 | 63.96 | – | 70.00 | – | 36.25 | – |
> | | LT | 62.31 (−1.7) | +60.7 | 73.33 (+3.3) | +73.4 | 37.03 (+0.8) | +71.1 |

---

> ### Author Response · Authors · 2025-11-21
>
> __Reasoning domains scopes__
>
> Thank you for raising this interesting point. Our main experiments focus on domains where previous work has shown that reasoning models exhibit stable and well-understood behavior (i.e., scientific, mathematical, and algorithmic tasks), and thus where these models are mainly applied to.
>
> However, we agree that assessing whether our latent-trajectory signals also generalize to less convergent, more “divergent” forms of reasoning is an interesting research question. To directly analyse this, we expanded our analysis to include tasks that require more commonsense-based, creative, narrative, or socially grounded forms of reasoning. Specifically, we evaluated our metrics on two BIG-Bench [1] tasks where existing reasoning models have not saturated performance and where the required cognitive processes differ substantively from the domains tested in the main paper:
>
> - Understanding Fables (https://github.com/google/BIG-bench/tree/main/bigbench/benchmark_tasks/understanding_fables): targeting narrative abstraction, commonsense, and creativity;
> - Social IQa (https://github.com/google/BIG-bench/tree/main/bigbench/benchmark_tasks/social_iqa): targeting social reasoning, emotional inference, and context-sensitive commonsense.
>
> Despite the very different nature of these problems, we observe that our LT signals remain significantly predictive of answer correctness in both settings. The ROC-AUC values for Net, Cumulative, and Aligned Change remain well above chance and are comparable to those reported for the main benchmarks (see Table below). These results suggest that the temporal structure we capture in latent space reflects a broader phenomenon in reasoning-oriented model behavior, not one limited to strictly “linear” or highly convergent problem-solving tasks.
>
> __Table: ROC-AUC scores for segment size = 100__
> | **Model**               | **Change**   | **AIME 2025** | **GPQA** | **TSP** | **BB Fables** | **BB Social** |
> |-------------------------|--------------|---------------|----------|---------|----------------|---------------|
> | Deepseek-R1-Qwen14B     | Net          | 0.696         | 0.693    | 0.636   | 0.693          | 0.619         |
> | Deepseek-R1-Qwen14B     | Cumulative   | 0.732         | 0.710    | 0.685   | 0.659          | 0.618         |
> | Deepseek-R1-Qwen14B     | Aligned      | 0.720         | 0.709    | 0.570   | 0.674          | 0.623         |
> | Phi4-Reasoning-Plus     | Net          | 0.781         | 0.752    | 0.708   | 0.777          | 0.713         |
> | Phi4-Reasoning-Plus     | Cumulative   | 0.802         | 0.749    | 0.786   | 0.775          | 0.714         |
> | Phi4-Reasoning-Plus     | Aligned      | 0.797         | 0.752    | 0.718   | 0.777          | 0.714         |
> | Qwen3-14B               | Net          | 0.937         | 0.665    | 0.667   | 0.839          | 0.696         |
> | Qwen3-14B               | Cumulative   | 0.947         | 0.687    | 0.713   | 0.844          | 0.683         |
> | Qwen3-14B               | Aligned      | 0.910         | 0.689    | 0.531   | 0.845          | 0.690         |
> | Deepseek-R1-Llama8B     | Net          | 0.888         | 0.633    | 0.514   | 0.653          | 0.595         |
> | Deepseek-R1-Llama8B     | Cumulative   | 0.894         | 0.628    | 0.582   | 0.651          | 0.587         |
> | Deepseek-R1-Llama8B     | Aligned      | 0.900         | 0.642    | 0.525   | 0.657          | 0.590         |
> | Deepseek-R1-Qwen32B     | Net          | 0.805         | 0.705    | 0.662   | 0.756          | 0.639         |
> | Deepseek-R1-Qwen32B     | Cumulative   | 0.814         | 0.691    | 0.690   | 0.732          | 0.629         |
> | Deepseek-R1-Qwen32B     | Aligned      | 0.821         | 0.712    | 0.579   | 0.746          | 0.639         |
>
>
> We believe these metrics remain predictive even in more “creative’’ or less linear reasoning tasks because they quantify relative differences between correct and incorrect trajectories rather than assuming any particular structure of the task. Even if a domain naturally produces larger or more exploratory latent movements overall, correct trajectories still tend to show more consistent and directed progression toward their final internal state, whereas incorrect ones typically wander more or exhibit less alignment. Thus, the absolute scale of representational change may vary across domains, but the contrast between successful and unsuccessful paths remains robust, which is what the LT signals capture.
>
> We include the full scores and distribution plots for these additional tasks in the revised appendix. While we agree that understanding how these signals behave in highly divergent or exploratory reasoning tasks is an important direction for future research, our extended experiments provide initial evidence that the metrics generalize beyond quantitative or strictly deductive reasoning domains.
>
> __References__
>
> [1] Srivastava et al. (2022), Beyond the Imitation Game (BIG-Bench), arXiv:2210.09261.

---

> > ### Author Response · Authors · 2025-11-21
> > **Part 3**
> >
> > __Prior work discussion__
> >
> > Thank you for pointing out this important prior work. We include a discussion of how our methods and findings position themselves with respect to these studies in the “Related Work” section.
> >
> > __Finer-grained tracking of hidden states__
> >
> > Thank you for this suggestion. We add to our appendix results showing that ROC-AUC scores remain significant across segment length choices:
> >
> > **Table: AUC–ROC results across segment lengths.**
> >
> > | **Model** | **Dataset** | **Net Change 100** | **Net Change 300** | **Net Change 500** | **Net Change 700** | **Cum. Change 100** | **Cum. Change 300** | **Cum. Change 500** | **Cum. Change 700** | **Aligned 100** | **Aligned 300** | **Aligned 500** | **Aligned 700** |
> > |-----------|-------------|--------------------|---------------------|---------------------|---------------------|----------------------|----------------------|----------------------|----------------------|------------------|------------------|------------------|------------------|
> > | **R1-D** | GPQA | 0.69 | 0.69 | 0.69 | 0.68 | 0.71 | 0.70 | 0.69 | 0.68 | 0.71 | 0.68 | 0.67 | 0.66 |
> > | | AIME2025 | 0.70 | 0.74 | 0.76 | 0.76 | 0.73 | 0.72 | 0.70 | 0.71 | 0.72 | 0.75 | 0.76 | 0.74 |
> > | | TSP | 0.64 | 0.61 | 0.64 | 0.66 | 0.69 | 0.69 | 0.69 | 0.69 | 0.57 | 0.64 | 0.66 | 0.68 |
> > | **Phi4R+** | GPQA | 0.75 | 0.74 | 0.74 | 0.75 | 0.75 | 0.74 | 0.74 | 0.74 | 0.75 | 0.74 | 0.74 | 0.73 |
> > | | AIME2025 | 0.78 | 0.77 | 0.75 | 0.79 | 0.80 | 0.79 | 0.79 | 0.78 | 0.80 | 0.77 | 0.76 | 0.77 |
> > | | TSP | 0.71 | 0.67 | 0.62 | 0.71 | 0.79 | 0.78 | 0.78 | 0.78 | 0.72 | 0.68 | 0.73 | 0.75 |
> > | **Qwen3** | GPQA | 0.67 | 0.67 | 0.67 | 0.65 | 0.69 | 0.67 | 0.66 | 0.65 | 0.69 | 0.66 | 0.63 | 0.62 |
> > | | AIME2025 | 0.94 | 0.93 | 0.92 | 0.92 | 0.95 | 0.95 | 0.95 | 0.95 | 0.91 | 0.93 | 0.91 | 0.92 |
> > | | TSP | 0.67 | 0.63 | 0.64 | 0.63 | 0.71 | 0.74 | 0.75 | 0.76 | 0.53 | 0.65 | 0.68 | 0.70 |
> >
> >
> > We also add figures showing that LT signals significantly distinguish accurate and inaccurate responses across step choices early in the trace (see Table below, now also added to appendix).
> >
> > **Figure:** Classifier accuracies across early reasoning segments.
> >
> > ---
> >
> > **Table: Classifier accuracy across early reasoning segments (1–5). Each segment = 500 tokens.**
> >
> > | **Model** | **Dataset** | **Seg 1** | **Seg 2** | **Seg 3** | **Seg 4** | **Seg 5** |
> > |-----------|-------------|-----------|-----------|-----------|-----------|-----------|
> > | **R1-D** | GPQA | 0.60 | 0.60 | 0.59 | 0.59 | 0.62 |
> > | | AIME2025 | 0.63 | 0.53 | 0.60 | 0.63 | 0.57 |
> > | | TSP | 0.29 | 0.32 | 0.29 | 0.26 | 0.29 |
> > | **Phi4R+** | GPQA | 0.72 | 0.68 | 0.70 | 0.72 | 0.69 |
> > | | AIME2025 | 0.83 | 0.83 | 0.87 | 0.83 | 0.80 |
> > | | TSP | 0.40 | 0.42 | 0.41 | 0.46 | 0.41 |
> > | **Qwen3** | GPQA | 0.65 | 0.66 | 0.63 | 0.66 | 0.63 |
> > | | AIME2025 | 0.73 | 0.77 | 0.77 | 0.73 | 0.73 |
> > | | TSP | 0.33 | 0.39 | 0.34 | 0.38 | 0.35 |
> >
> >
> > In addition, our appendix also includes figures of LT-signals distribution across layers of the models.

---

> ### Comment · Reviewer_SL11 · 2025-11-28
>
> Thank you for addressing my points and presenting the additional results.
>
> These new results show that the method seems to be fairly robust with respect to the thresholding hyperparameter, and different types of reasoning.
>
> While I agree with Reviewer NXfb that a detailed, perhaps theoretical explanation of _why_ these latent state signals are predictive of accuracy would be desirable, I regard the paper's contribution with its current scope as strong enough. However, I believe there could be interesting follow-up work leaning more towards the interpretability side.
>
> In light of the clarifications and the robust empirical evidence, I will increase my score.

---

### Official Review · Reviewer_AG1e · 2025-11-01

**Soundness:** 3
**Presentation:** 4
**Contribution:** 3
**Rating:** 6
**Confidence:** 4

**Summary:**

This paper introduces a method for predicting the correctness of a reasoning path by analyzing the direction and magnitude of change in latent vectors over the temporal dimension. The goal is to infer the quality of a reasoning path in advance by examining its latent dynamics during the intermediate generation process. Experiments demonstrate that the proposed Latent-Trajectory (LT) signals correlate with final answer accuracy. Furthermore, the paper shows that these LT signals can be practically applied for early answer selection and path pruning, significantly reducing the number of tokens generated while maintaining, and in some cases improving, overall accuracy.

**Strengths:**

1. Training-free method: The core method is training-free and does not rely on semantic understanding. It provides a novel way to discriminate reasoning quality based purely on the internal dynamics of the model's hidden states, making it a lightweight and potentially generalizable approach.

2. Positive experimental results: The experiments successfully show a positive correlation between the proposed LT signals and final answer accuracy. More importantly, the paper demonstrates the practical utility of these signals in two efficiency-oriented applications (Early Answer Selection and Early Path Selection), achieving notable reductions in number of tokens while preserving or even slightly improving accuracy .

3. Novel methodological perspective: The central idea of analyzing the temporal evolution of latent states is a novel contribution. It offers a new lens for understanding the reasoning process.

**Weaknesses:**

1. Moderate correlation and robustness: The reported correlations (Spearman's r in Table 3) are moderate, often falling in the 0.3-0.7 range. While the authors show this is sufficient for the tested datasets, this moderate correlation suggests the signal is somewhat noisy and raises concerns about the method's robustness when applied to different models, domains, or tasks.

2. Heavy reliance on unjustified hyperparameters: The methodology is built on several ad-hoc, empirical settings that lack strong theoretical or intuitive justification. For example, segmentation (k=500), averaging (across-layer), or checkpoint (t=2000) in early pruning experiment. This work does not provide a guide on how to set this parameters when applied to other tasks.

3. Missing Comparison to Key Related Work: A critical weakness is the lack of experimental comparison to highly relevant, concurrent work. Specifically, the paper does not compare ST-BoN [1], which shares an identical motivation (efficient Best-of-N) but uses a competing approach (spatial/layer-wise trajectory analysis). Given the strong overlap in goals, a direct comparison is necessary to properly situate this work's contribution.

Reference:
[1] Wang Y, Zhang P, Huang S, et al. Sampling-efficient test-time scaling: Self-estimating the best-of-n sampling in early decoding[J]. arXiv preprint arXiv:2503.01422, 2025.

4. Inefficient experimental setup for Early Selection: The "Early Answer Selection" experiment relies on a serial generation strategy (generating samples one by one). This approach discards the parallel processing capabilities of modern inference infrastructure (i.e., batch decoding) and is likely highly inefficient in real-world applications.

**Questions:**

1. For the "Early Answer Selection" experiment, how does the LT signal-based stopping rule compare to simpler serial baselines? For instance, what is the accuracy if one simply stops after the first sample (k=1) or the second (k=2)? A key concern is that the tested datasets might have low output variance, and a simple k=1 or k=2 rule might achieve similar results without the complexity of calculating LT signals.

2. The method's reliance on calibration on a labeled (correct/incorrect) dataset seems to limit its application. How do the authors envision this method being practiced in more general, open-ended scenarios (e.g., creative writing, general instruction-following) where there is no single ground truth and "correct/incorrect" labels for calibration are unavailable?

---

> ### Author Response · Authors · 2025-11-21
> **Part 1**
>
> Thank you for the thorough review and constructive feedback. We appreciate your positive comments on our method being training-free, the novel perspective of analyzing latent temporal dynamics, and the practical utility shown in the early selection and pruning experiments. Below, we address each of the raised concerns and clarify the methodological choices, additional experiments, and generalization analyses included in the revised manuscript.
>
>
> __Moderate correlation and robustness__
>
> Thank you for raising this point. While the Spearman values fall in the 0.3–0.7 range, this level of correlation is expected for internal signals based on complex and varying model states. Importantly, these values are notably stronger and more stable than commonly used baselines such as logit margin, entropy, perplexity, or cross-layer signals (full comparisons already included in Appendix A). Even if moderate, LT signals show consistent and meaningful correlations with correctness across all evaluated model–dataset pairs. These signals applied in practice translate into substantial gains, as demonstrated in Tables 1–2 in the paper, which is a better indicator of usefulness than correlation.
>
> To further assess robustness, we expanded our evaluation beyond the original three models and reasoning domains. The revised manuscript will include results on additional model classes and sizes (specifically DeepSeek-R1-Distill-Llama-8B and DeepSeek-R1-Distill-Qwen-32B) as well as on more diverse reasoning domains such as narrative abstraction and social commonsense (BIG-Bench Understanding Fables and Social IQa). Across these extended settings, LT signals remain reliably predictive (average value of 0.72 and 0.09 standard deviation; see Table below), reinforcing that the observed phenomena are not restricted to a narrow architecture or task type.
>
>
> __Table: ROC-AUC scores for segment size = 100__
> | **Model**               | **Change**   | **AIME 2025** | **GPQA** | **TSP** | **BB Fables** | **BB Social** |
> |-------------------------|--------------|---------------|----------|---------|----------------|---------------|
> | Deepseek-R1-Qwen14B     | Net          | 0.696         | 0.693    | 0.636   | 0.693          | 0.619         |
> | Deepseek-R1-Qwen14B     | Cumulative   | 0.732         | 0.710    | 0.685   | 0.659          | 0.618         |
> | Deepseek-R1-Qwen14B     | Aligned      | 0.720         | 0.709    | 0.570   | 0.674          | 0.623         |
> | Phi4-Reasoning-Plus     | Net          | 0.781         | 0.752    | 0.708   | 0.777          | 0.713         |
> | Phi4-Reasoning-Plus     | Cumulative   | 0.802         | 0.749    | 0.786   | 0.775          | 0.714         |
> | Phi4-Reasoning-Plus     | Aligned      | 0.797         | 0.752    | 0.718   | 0.777          | 0.714         |
> | Qwen3-14B               | Net          | 0.937         | 0.665    | 0.667   | 0.839          | 0.696         |
> | Qwen3-14B               | Cumulative   | 0.947         | 0.687    | 0.713   | 0.844          | 0.683         |
> | Qwen3-14B               | Aligned      | 0.910         | 0.689    | 0.531   | 0.845          | 0.690         |
> | Deepseek-R1-Llama8B     | Net          | 0.888         | 0.633    | 0.514   | 0.653          | 0.595         |
> | Deepseek-R1-Llama8B     | Cumulative   | 0.894         | 0.628    | 0.582   | 0.651          | 0.587         |
> | Deepseek-R1-Llama8B     | Aligned      | 0.900         | 0.642    | 0.525   | 0.657          | 0.590         |
> | Deepseek-R1-Qwen32B     | Net          | 0.805         | 0.705    | 0.662   | 0.756          | 0.639         |
> | Deepseek-R1-Qwen32B     | Cumulative   | 0.814         | 0.691    | 0.690   | 0.732          | 0.629         |
> | Deepseek-R1-Qwen32B     | Aligned      | 0.821         | 0.712    | 0.579   | 0.746          | 0.639         |

---

> > ### Author Response · Authors · 2025-11-21
> > **Part 2**
> >
> > __Hyperparameter selection__
> >
> > Thank you for highlighting this point. We expanded our experiments to directly address these concerns, and our findings confirm that the method is not sensitive to any single hyperparameter choice:
> >
> > - _Segmentation length_: As shown in the table below (and included in the appendix), we found that results remain stable across segment length choice (k={100,300,500,700}). For each LT signal, the fluctuations across segment lengths are overall small (with an average variation across segment lengths of 0.01-0.05 for a model-dataset pair).
> >
> >
> > | **Model** | **Dataset** | **Net Change 100** | **Net Change 300** | **Net Change 500** | **Net Change 700** | **Cum. Change 100** | **Cum. Change 300** | **Cum. Change 500** | **Cum. Change 700** | **Aligned 100** | **Aligned 300** | **Aligned 500** | **Aligned 700** |
> > |-----------|-------------|--------------------|---------------------|---------------------|---------------------|----------------------|----------------------|----------------------|----------------------|------------------|------------------|------------------|------------------|
> > | **R1-D** | GPQA | 0.69 | 0.69 | 0.69 | 0.68 | 0.71 | 0.70 | 0.69 | 0.68 | 0.71 | 0.68 | 0.67 | 0.66 |
> > | | AIME2025 | 0.70 | 0.74 | 0.76 | 0.76 | 0.73 | 0.72 | 0.70 | 0.71 | 0.72 | 0.75 | 0.76 | 0.74 |
> > | | TSP | 0.64 | 0.61 | 0.64 | 0.66 | 0.69 | 0.69 | 0.69 | 0.69 | 0.57 | 0.64 | 0.66 | 0.68 |
> > | **Phi4R+** | GPQA | 0.75 | 0.74 | 0.74 | 0.75 | 0.75 | 0.74 | 0.74 | 0.74 | 0.75 | 0.74 | 0.74 | 0.73 |
> > | | AIME2025 | 0.78 | 0.77 | 0.75 | 0.79 | 0.80 | 0.79 | 0.79 | 0.78 | 0.80 | 0.77 | 0.76 | 0.77 |
> > | | TSP | 0.71 | 0.67 | 0.62 | 0.71 | 0.79 | 0.78 | 0.78 | 0.78 | 0.72 | 0.68 | 0.73 | 0.75 |
> > | **Qwen3** | GPQA | 0.67 | 0.67 | 0.67 | 0.65 | 0.69 | 0.67 | 0.66 | 0.65 | 0.69 | 0.66 | 0.63 | 0.62 |
> > | | AIME2025 | 0.94 | 0.93 | 0.92 | 0.92 | 0.95 | 0.95 | 0.95 | 0.95 | 0.91 | 0.93 | 0.91 | 0.92 |
> > | | TSP | 0.67 | 0.63 | 0.64 | 0.63 | 0.71 | 0.74 | 0.75 | 0.76 | 0.53 | 0.65 | 0.68 | 0.70 |
> >
> >
> > - _Checkpoint selection_: As shown in the below (and included in the appendix), LT-based accuracy remains significant and stable across early checkpoint choices. The average variation across segments is only 0.05.
> >
> >
> > | **Model** | **Dataset** | **500 tokens** | **1000 tokens** | **1500 tokens** | **2000 tokens** | **2500 tokens** |
> > |-----------|-------------|-----------|-----------|-----------|-----------|-----------|
> > | **R1-D** | GPQA | 0.60 | 0.60 | 0.59 | 0.59 | 0.62 |
> > | | AIME2025 | 0.63 | 0.53 | 0.60 | 0.63 | 0.57 |
> > | | TSP | 0.29 | 0.32 | 0.29 | 0.26 | 0.29 |
> > | **Phi4R+** | GPQA | 0.72 | 0.68 | 0.70 | 0.72 | 0.69 |
> > | | AIME2025 | 0.83 | 0.83 | 0.87 | 0.83 | 0.80 |
> > | | TSP | 0.40 | 0.42 | 0.41 | 0.46 | 0.41 |
> > | **Qwen3** | GPQA | 0.65 | 0.66 | 0.63 | 0.66 | 0.63 |
> > | | AIME2025 | 0.73 | 0.77 | 0.77 | 0.73 | 0.73 |
> > | | TSP | 0.33 | 0.39 | 0.34 | 0.38 | 0.35 |
> >
> >
> > Regarding the averaging scheme, we decided to aggregate signals across layers because, as shown in the distribution plots in Appendix B, the specific layers in which the strongest difference between accurate and inaccurate responses is found to vary across models. Averaging signals across layers thus improves the generalizability of the method across architectures, without having to pick model-specific layers.
> >
> > Moreover, our claim is not that the specific inference-time implementations we present are the only or optimal way to exploit LT signals. Instead, the contribution of the paper is to show that LT signals contain reliable information that can be leveraged at inference time, and we provide a straightforward baseline method for doing so. As we note in the limitations section, developing more sophisticated mechanisms for leveraging LT signals is an important direction for future work. We will make this point more explicit in the revised manuscript.

---

> > > ### Author Response · Authors · 2025-11-21
> > > **Part 3**
> > >
> > > __Comparison to Wang et al. 2025__
> > >
> > > Thank you for pointing out this relevant concurrent work. We have now added a direct comparison to ST-BoN (Wang et al., 2025) using our early-trace experimental setting.
> > >
> > > We present the comparison below and will include it in Table 2 of our revised manuscript. We found that across nearly all model–dataset settings, LT-based early selection matches or exceeds the accuracy of ST-BoN while consistently delivering comparable or greater token savings. While ST-BoN exhibits substantial variability (including drops of up to 4%), LT’s deviations are small and its improvements are more uniform, frequently achieving the highest accuracy. Across models and datasets, LT delivers more than double the average accuracy gain of ST-BoN (+2.48% vs +1.12%). Token savings for LT and ST-BoN are similar, but LT is slightly higher on average (+61.2% vs +59.9%).
> > >
> > > In addition, LT remains computationally cheaper, since it operates on the latent trajectory of a single sample rather than computing pairwise distances across multiple samples as required by ST-BoN.
> > >
> > >
> > > | **Model** | **Strategy** | **GPQA Accuracy (% / Δ%)** | **GPQA Saved Tokens (Δ%)** | **AIME2025 Accuracy (% / Δ%)** | **AIME2025 Saved Tokens (Δ%)** | **TSP Accuracy (% / Δ%)** | **TSP Saved Tokens (Δ%)** |
> > > |-----------|--------------|-----------------------------|------------------------------|----------------------------------|----------------------------------|-----------------------------|-----------------------------|
> > > | **R1-D** | Maj@5 | 59.90 | – | 56.67 | – | 27.50 | – |
> > > | | ST-BoN | 58.88 (−1.02) | +45.3 | 63.33 (+6.7) | +49.7 | **28.75** (+1.2) | +62.8 |
> > > | | LT | **59.39** (−0.5) | +48.9 | 63.33 (+6.7) | +50.1 | 26.25 (−1.3) | +62.5 |
> > > | **Phi4R+** | Maj@5 | 70.20 | – | 80.00 | – | 41.25 | – |
> > > | | ST-BoN | 66.16 (−4.0) | +62.3 | 76.67 (−3.3) | +67.4 | 42.50 (+1.25) | +71.4 |
> > > | | LT | **72.22** (+2.0) | +64.7 | **83.33** (+3.3) | +67.3 | **45.63** (+4.4) | +71.7 |
> > > | **Qwen3** | Maj@5 | 63.96 | – | 70.00 | – | 36.25 | – |
> > > | | ST-BoN | 63.63 (+2.54) | +46.8 | **76.67** (+6.7) | +67.4 | 36.25 (+0.0) | +65.8 |
> > > | | LT | **66.50** (+2.5) | +51.0 | 73.33 (+3.3) | +69.1 | **38.13** (+1.9) | +65.7 |

---

> > > > ### Author Response · Authors · 2025-11-21
> > > > **Part 4**
> > > >
> > > > __Choice of serial generation__
> > > >
> > > > Although this is a valid concern, serial generation does not imply inefficient execution. In practice, serial sampling can be implemented using a batching strategy in which multiple different problems are decoded in parallel, effectively utilizing hardware parallelization strategies. When a particular problem needs an additional sample, it is simply added to a later batch alongside other questions at their own decoding steps. This means that, although sampling is sequential for each individual question, the GPU remains fully utilized because batches can be formed across many questions. While we acknowledge that this could potentially affect latency, our time savings from earlier answer selection help avoid worst-case complexity. We thus provide early-in-the-trace experiments (Section 5.3) that score partial trajectories well before completion, reducing or eliminating the need for full serial sampling if that is preferred. These results further mitigate the concern about inference inefficiency.
> > > >
> > > > Moreover, our main goal was not to propose an inference-time deployment strategy but to determine and characterize whether latent-trajectory signals contain useful information about the quality of reasoning traces. Even if some deployment scenarios prefer parallel decoding, understanding the informational content of these internal signals remains important.

---

> > > > > ### Author Response · Authors · 2025-11-21
> > > > > **Part 5**
> > > > >
> > > > > __Serial baseline comparison__
> > > > >
> > > > > Thank you for raising this point. Following your suggestion, we added a sequential baseline strategy that stops after the first generated sample (“First Sample”). As shown in the table below (now included in the Appendix), this heuristic is occasionally competitive, but it is also unreliable: while it matches LT performance in a few cases and is slightly higher in a small number of settings, it also suffers substantial drops in others (e.g., −6 to −10 points on some complex datasets). This variability confirms that outcome quality for early samples is highly inconsistent across tasks and models, meaning there is no dependable way to know when the first or second sample will be much worse.
> > > > >
> > > > > In contrast, LT-based selection offers a principled and stable alternative. It uses information already produced during decoding (so computational cost is essentially unchanged), and it delivers consistently strong performance, often yielding notable gains (+5–9 points) on more challenging datasets. Even in cases where LT is slightly below the first-sample heuristic, the differences are small, whereas the improvements in the opposite direction can be large.
> > > > >
> > > > > Overall, while “k=1” can occasionally appear competitive, its behavior is fragile and dataset-dependent. LT signals provide a predictable, task-robust, and low-cost early-stopping mechanism that avoids the unpredictability inherent to fixed-k serial heuristics.
> > > > >
> > > > > **Table: Accuracy with Latent-Trajectory (LT) signals.  All accuracies in percent. “First Sample” is the baseline; Δ is the gain relative to this baseline.**
> > > > >
> > > > > | **Model** | **Strategy** | **GPQA (Acc / Δ)** | **AIME2025 (Acc / Δ)** | **TSP (Acc / Δ)** |
> > > > > |-----------|--------------|---------------------|--------------------------|---------------------|
> > > > > | **R1-D** | First Sample | 60.41 | 56.67 | 28.13 |
> > > > > | | LT–Net | 61.10 (+0.69) | 61.90 (+5.23) | 28.60 (+0.47) |
> > > > > | | LT–Cumulative | 62.10 (+1.69) | 58.70 (+2.03) | 30.90 (+2.77) |
> > > > > | | LT–Aligned | 61.10 (+0.69) | 60.30 (+3.63) | 29.50 (+1.37) |
> > > > > | **Phi4R+** | First Sample | 69.70 | 73.33 | 43.13 |
> > > > > | | LT–Net | 68.80 (−0.90) | 79.40 (+6.07) | 42.30 (−0.83) |
> > > > > | | LT–Cumulative | 69.60 (−0.10) | 81.00 (+7.67) | 44.40 (+1.27) |
> > > > > | | LT–Aligned | 69.60 (−0.10) | 82.50 (+9.17) | 44.10 (+0.97) |
> > > > > | **Qwen3** | First Sample | 63.13 | 76.67 | 36.25 |
> > > > > | | LT–Net | 63.70 (+0.57) | 79.40 (+2.73) | 35.40 (−0.85) |
> > > > > | | LT–Cumulative | 63.30 (+0.17) | 84.10 (+7.43) | 36.30 (+0.05) |
> > > > > | | LT–Aligned | 64.20 (+1.07) | 80.90 (+4.23) | 37.80 (+1.55) |

---

> > > > > > ### Author Response · Authors · 2025-11-21
> > > > > > **Part 6**
> > > > > >
> > > > > > __Labeled calibration__
> > > > > >
> > > > > > Thank you for raising this point. One way of addressing datasets without correct/incorrect labels is to determine thresholds based on other labeled datasets. To examine this possibility, we conducted new experiments that show that threshold calibration generalizes robustly across datasets and domains. Specifically, in our cross-dataset transfer experiment, we show how a percentile threshold learned on one dataset (e.g., GPQA) transfers effectively to unseen datasets such as AIME 2025 and TSP for the same model. We evaluated all train-test dataset combinations, and in every case, the transferred percentile achieved accuracy close to the within-dataset setting while maintaining the same 20–70% token savings (see Table below, now included in the Appendix). This demonstrates that, in practice, threshold calibration does not need to be performed on the target domain, enabling use in settings without correctness labels.
> > > > > >
> > > > > > **Table: Cross-dataset threshold generalization.**
> > > > > >
> > > > > > | **Model** | **Strategy** | **GPQA Acc (avg % / ΔAcc)** | **GPQA Samples (avg / ΔTok%)** | **AIME2025 Acc (avg % / ΔAcc)** | **AIME2025 Samples (avg / ΔTok%)** | **TSP Acc (avg % / ΔAcc)** | **TSP Samples (avg / ΔTok%)** |
> > > > > > |-----------|--------------|------------------------------|----------------------------------|-----------------------------------|--------------------------------------|------------------------------|---------------------------------|
> > > > > > | **R1-D** | MV@5 | 59.90 | 5.00 | 56.67 | 5.00 | 27.50 | 5.00 |
> > > > > > | | LT–Combined | 60.40 (+0.5) | 2.11 (+41.5%) | 60.00 (+3.3) | 1.40 (+60.3%) | 28.45 (+1.0) | 1.59 (+67.3%) |
> > > > > > | **Phi4R+** | MV@5 | 70.20 | 5.00 | 80.00 | 5.00 | 41.25 | 5.00 |
> > > > > > | | LT–Combined | 69.20 (−1.0) | 2.64 (+25.5%) | 80.00 (+0.0) | 2.87 (+23.1%) | 41.85 (+0.6) | 2.52 (+45.2%) |
> > > > > > | **Qwen3** | MV@5 | 63.96 | 5.00 | 70.00 | 5.00 | 36.25 | 5.00 |
> > > > > > | | LT–Combined | 63.60 (−0.4) | 2.08 (+45.4%) | 76.65 (+6.7) | 2.18 (+40.5%) | 35.90 (−0.3) | 1.29 (+73.6%) |
> > > > > >
> > > > > >
> > > > > > To further support this type of generalization, we conducted a leave-one-out early-trace selection experiment: for each model, we trained a Random Forest classifier on LT signals from two datasets and evaluated it on the third completely unseen dataset. This held-out testing setup shows that LT-based early selection maintains competitive accuracy and achieves 60–75% token savings, even though the selector has never seen labels from the held-out domain (see Table below, now included in Appendix). This provides evidence that LT-based calibration can transfer even across differing reasoning distributions.
> > > > > >
> > > > > > **Table: Cross-dataset generalization early in the trace.**
> > > > > >
> > > > > > | **Model** | **Strategy** | **GPQA Accuracy (% / Δ%)** | **GPQA Saved Tokens (Δ%)** | **AIME2025 Accuracy (% / Δ%)** | **AIME2025 Saved Tokens (Δ%)** | **TSP Accuracy (% / Δ%)** | **TSP Saved Tokens (Δ%)** |
> > > > > > |-----------|--------------|-----------------------------|------------------------------|----------------------------------|----------------------------------|-----------------------------|-----------------------------|
> > > > > > | **R1-D** | Maj@5 | 59.90 | – | 56.67 | – | 27.50 | – |
> > > > > > | | LT | 60.91 (+1.0) | +59.7 | 56.67 (+0.0) | +59.2 | 30.78 (+3.3) | +69.4 |
> > > > > > | **PhiR+** | Maj@5 | 70.20 | – | 80.00 | – | 41.25 | – |
> > > > > > | | LT | 69.07 (−1.1) | +70.4 | 82.50 (+2.5) | +71.3 | 40.00 (−1.3) | +75.1 |
> > > > > > | **Qwen3** | Maj@5 | 63.96 | – | 70.00 | – | 36.25 | – |
> > > > > > | | LT | 62.31 (−1.7) | +60.7 | 73.33 (+3.3) | +73.4 | 37.03 (+0.8) | +71.1 |
> > > > > >
> > > > > >
> > > > > > In addition, we expanded our evaluation to broader reasoning tasks outside the structured domains. Beyond GPQA, AIME 2025, and TSP, we tested LT signals on two BIG-Bench tasks requiring narrative abstraction, creativity, and social commonsense: Understanding Fables and Social IQa. Despite the shift toward more divergent reasoning styles, LT signals remained strongly predictive of correctness (ROC-AUC values well above chance and comparable to those in structured tasks). These results (included in response to “Moderate correlation and robustness” concern) show that LT generalizes not only across datasets but also across qualitatively different reasoning processes.
> > > > > >
> > > > > > Taken together, these experiments indicate that LT can be applied in settings without strict correctness labels. The threshold need not be calibrated on the target domain, and LT signals remain informative even in tasks that do not have a single, well-defined correct answer.
> > > > > >
> > > > > > We also note that reasoning models are primarily evaluated and applied in domains with objective correctness (math, science, algorithmic reasoning), which aligns with our main experimental setting.

---

> > > > > > > ### Comment · Reviewer_AG1e · 2025-11-24
> > > > > > >
> > > > > > > Thank you for your detailed replies and work on rebuttle. The experiments on robustness and hyperparameter resolved my concerns. Comparative and supplementary experiments are also added and the results demonstrate the effectiveness of the proposed methods. I have no further questions.

---

> > > > > > > > ### Author Response · Authors · 2025-11-24
> > > > > > > > **Thank you**
> > > > > > > >
> > > > > > > > Thank you very much for your rapid follow-up and for letting us know that your concerns have been resolved. We appreciate your time and engagement.
> > > > > > > >
> > > > > > > > Since the issues you previously raised were central to your score, we would appreciate it if you could consider updating your score. Please let us know if there are any remaining points we can clarify/expand to support a fair update.

---

### Meta-Review · Area_Chair_nAJu · 2026-01-12

**Summary:**

This submission introduces latent trajectory signals to quantify changes in the model’s hidden states during inference and assess the quality of reasoning paths.
The reviewers raised the following concerns.

* Reviewers AG1e and NXfb questioned the robustness of the proposed method across datasets and architectures. The authors addressed this concern by expanding the evaluations to include models of different sizes and from a different architectural family. The authors also added comparisons against a listed concurrent work.

* Reviewers AG1e and SL11 asked whether the proposed approach requires non-standard and dataset-specific hyperparameter tuning. In the revision, the authors showed via experiments that results are not HP-sensitive.

* Reviewer SL11 asked whether the benefit of a “direct” trace is task-specific, which prompted the authors to include experiments on more open-ended tasks.

* Reviewer NXfb requested comparison against alternative approaches to improve reasoning such as RL, and causal interventions on the reasoning traces. The authors partly addressed these concerns by clarifying the scope of the work.

Overall, the authors provided a diligent rebuttal that addressed most of the major concerns. The area chair hence recommends acceptance and encourages the authors to carefully revise the manuscript.

**Reviewer Concerns:**

See above.

**Reviewer Scores:**

Reviewers AG1e and SL11 gave a score of borderline accept, and the authors addressed all of their major concerns in the rebuttal. Reviewer NXfb gave a negative score of 2, and the listed concerns are only partially addressed; the area chair believes that an improved score to 4 seems likely.

---

### Decision · Program_Chairs · 2026-01-26

Accept (Poster)